# Synthetic Pharmacotherapy for Systemic Lupus Erythematosus: Potential Mechanisms of Action, Efficacy, and Safety

**DOI:** 10.3390/medicina59010056

**Published:** 2022-12-27

**Authors:** Angélica María Téllez Arévalo, Abraham Quaye, Luis Carlos Rojas-Rodríguez, Brian D. Poole, Daniela Baracaldo-Santamaría, Claudia M. Tellez Freitas

**Affiliations:** 1Department of Physiological Sciences, School of Medicine, Pontificia Universidad Javeriana, Carrera 7 No. 40–62, Bogotá 110231, Colombia; 2Department of Microbiology and Molecular Biology, Brigham Young University, Provo, UT 84602, USA; 3Pharmacology Unit, Department of Biomedical Sciences, School of Medicine and Health Sciences, Universidad del Rosario, Bogotá 111221, Colombia; 4College of Dental Medicine, Roseman University of Health Sciences, South Jordan, UT 84095, USA

**Keywords:** immunosuppressant agent, glucocorticoid, antimalarial drug, efficacy, safety, systemic lupus erythematosus

## Abstract

The pharmacological treatment of systemic lupus erythematosus (SLE) aims to decrease disease activity, progression, systemic compromise, and mortality. Among the pharmacological alternatives, there are chemically synthesized drugs whose efficacy has been evaluated, but which have the potential to generate adverse events that may compromise adherence and response to treatment. Therapy selection and monitoring will depend on patient characteristics and the safety profile of each drug. The aim of this review is to provide a comprehensive understanding of the most important synthetic drugs used in the treatment of SLE, including the current treatment options (mycophenolate mofetil, azathioprine, and cyclophosphamide), review their mechanism of action, efficacy, safety, and, most importantly, provide monitoring parameters that should be considered while the patient is receiving the pharmacotherapy.

## 1. Introduction

The pharmacological treatment of systemic lupus erythematosus (SLE) aims to decrease disease activity, progression, systemic compromise, and mortality, thus improving patient quality of life [1]. However, all pharmacological alternatives can potentially precipitate adverse reactions, which is why risk management programs that evaluate the safety and efficacy of treatments must be an integral part of the selection and during therapy. A wide variety of drugs are available; therapy choice depends on many diverse factors, such as organ systems compromised, disease activity, previous therapy response, desire for parenthood, pregnancy or lactation, contraindications, and therapy adherence [2,3].

Non-adherence to drug therapy in SLE patients is a significant obstacle, as it ranges between 3–76% of patients and is associated with disease progression and increased morbimortality [4]. Education of all SLE patients at the time of diagnosis regarding the disease, the selected drugs, and non-adherence consequences is predicted to play a vital role in circumventing therapy non-adherence [3].

This section aims to provide a general description of chemically synthesized pharmacological agents for SLE treatment, emphasizing their safety margins and monitoring parameters.

## 2. Antimalarial Drugs

Chloroquine (CQ) and hydroxychloroquine (HCQ, a hydroxylated analog of CQ) are chemically synthesized disease-modifying drugs derived from alkaloids found in the cortex of *Chinchona officinalis* [5]. CQ and HCQ were synthesized in 1934 and 1950, respectively, and approved by the FDA for medical use in 1949 and 1955, respectively. Their anti-inflammatory and immunomodulatory effects underlie their effectiveness in treating SLE and other immunopathological diseases [6].

CQ and HCQ are lipophilic drugs that enter cells by simple diffusion. Their basic side chains concentrate in acidic organelles, such as endosomes, lysosomes, and Golgi vesicles, increasing the organelles’ pH and interfering with multiple cellular processes involved in innate and adaptive immunity [7]. The pH changes in lysosomes—an increase from ~4.7 to ~6 [8]—destabilize its membrane and promote the loss of lysosomal enzymes in the cytosol. The pH changes also impede lysosomal enzyme function, which impairs endolysosome cargo degradation in autophagy, endocytosis, and phagocytosis pathways, essential in antigen processing for presentation [6] (See Figure 1).

Other immunomodulatory mechanisms of CQ and HCQ include: (1) Interference with the activation and signaling of Toll-like receptors 7 and 9 present on endosomal surfaces, which are involved in inflammatory responses and production of co-stimulatory molecules that participate in antigen presentation [6]. (2) Interference of cyclic GMP/AMP synthase, an essential enzyme in the function of type I interferon and IL-1. (3) Inhibition of Phospholipase A2 (PLA2) [7]. (4) Downregulation of the synthesis of proinflammatory cytokines, such as IL-1, IL-6, TNF, and IFN-γ in T and B cells [6]. (5) Enhancement of nitric oxide production by endothelial cells, inhibition of platelet aggregation [9,10], reduced formation of aPL-β2GPI complexes in phospholipid bilayers, and restoration of the anticoagulant function of annexin 5, collectively leading to vascular protective effects [8,11].

### 2.1. Efficacy

Antimalarials are most efficacious in treating mucocutaneous and musculoskeletal SLE [12]. They are effective in flare prevention and reduction of disease activity and mortality [13,14]. Accordingly, non-adherence to HCQ treatment was found to increase the patient’s risk of flare-ups with lower complement C4 values, particularly those who only complied for less than a year [15]. CQ was shown in a randomized placebo-controlled trial of 24 patients to reduce disease progression rates, therapeutic glucocorticoid (GC) doses, and SLEDAI (Systemic Lupus Erythematosus Disease Activity Index) scores [16].

Antimalarials have also proved potent in managing lupus nephritis (LN), the commonest life-threatening complication of SLE [17]. A study showed that they improved the 12-month renal therapy response rates, reduced the risk of flares, and delayed the progression of renal insufficiency [18]. Additionally, they improve clinical outcomes in pregnant patients with SLE [18]. In a retrospective cohort study of 151 pregnant SLE patients, preeclampsia incidence was significantly lower, and neonatal weight was significantly greater, in patients treated with HCQ than control patients [19]. Furthermore, antimalarials reduced the risk of neonatal cardiac manifestations in pregnant SLE patients positive for anti-SSA/Ro antibodies [20].

Other benefits include a reduced risk of thromboembolism in patients with antiphospholipid antibodies [9], better glycemic control in SLE patients, and improved insulin sensitivity, thus decreasing the risk of developing diabetes [21,22]. The use of antimalarials leads to improved lipid profiles through reduced cholesterol synthesis and LDL receptor activity [7,22,23], improved bone density [24], and lower cancer risk [25].

### 2.2. Safety

HCQ and CQ have an established good safety profile and are usually well tolerated [7]. They are considered immunomodulatory, but not immunosuppressive, because their usage is not associated with an increased risk of infection or cancer [1]. Despite their ubiquitous clinical use in treating several inflammatory rheumatic diseases, antimalarials have an undetermined dose–response relationship and no defined minimum clinically efficacious dose. Hence, predicting dose-dependent side effects or toxicity is a challenge in clinical practice.

#### 2.2.1. Gastrointestinal Adverse Reactions

The main adverse reactions associated with the antimalarials are gastrointestinal effects, including nausea, vomiting, abdominal pain, and diarrhea. For better tolerance, it is recommended to take them with meals. Cases of elevated liver enzymes in liver function tests and fulminant liver failure have been described. Hence, cautious use in patients with liver disease, alcoholism, or known use of hepatotoxic drugs is necessary [26].

#### 2.2.2. Dermatologic Adverse Events

Generalized itching that responds poorly to antihistamine treatment, beginning a few hours after taking antimalarials, may occur. However, the itch usually resolves spontaneously within 72 h [8]. Furthermore, aquagenic pruritus has been described, but it is rare [27]. Another cutaneous manifestation is the loss of hair pigmentation [28] and hyperpigmentation; these are more frequent in patients with ecchymosis and those using antiplatelet agents or anticoagulants [29]. Still, other cutaneous reactions, such as DRESS (Drug Reaction with Eosinophilia and Systemic Symptoms), erythema multiforme, erythroderma, generalized exanthematous pustulosis, Stevens–Johnson syndrome, toxic epidermal necrolysis, or psoriasis exacerbations, may occur in the first days or weeks of treatment. These reactions strictly require treatment suspension [28].

#### 2.2.3. Ocular Toxicity

CQ and HCQ have an affinity for melanin, whereby prolonged exposure causes concentration in melanin-rich tissues, such as the skin and retina [30]. When accumulated in these cells, they interfere with lysosome function, inhibiting autophagy and stimulating lipofuscin accumulation, which results in toxicity to the photoreceptors and the retinal epithelial cells [6,31]. HCQ also inhibits the activity of the OATP1A2 (Organic Anion Transporter Polypeptide 1A2), involved in the recycling of all-trans-retinol in the retinal epithelial cells, an essential step in the visual cycle [32]. This interference causes retinopathy, the most severe complication of antimalarial treatment [30,32]. Risk factors for ocular toxicity are described in Table 1.

Clinical features include difficulty reading, scotomas, reduced visual acuity, altered color perception, and diminished peripheral and nocturnal vision [30,31,32]. In the fundoscopic examination, bulls-eye maculopathy can be visualized, a characteristic image caused by depigmentation of the retinal pigment epithelium of the macula with an unaffected central portion [31]. If exposure to the drug continues, retinal atrophy and retinitis pigmentosa may occur [33].

#### 2.2.4. Cardiotoxicity

Antimalarial-induced cardiomyopathy (AICM) is a rare complication secondary to myocardial lysosomal dysfunction and cation (Na, K, and Ca) current alterations in the heart’s electrical conduction system [34]. The features of AICM include hypertrophic and restrictive cardiomyopathy with or without conduction and rhythm abnormalities, bradycardia, tachycardia, T wave flattening, cQT interval prolongation [7,34], right bundle branch block, left anterior fascicular block, and complete atrioventricular block [35]. Due to cQT prolongation, pharmacological interactions that further increase the risk for ventricular arrhythmia must be avoided. Risk factors for AICM include advanced age, female sex, exposure for more than ten years, high daily dose per kg of body weight, high cumulative dose, preexistent cardiac disease, liver disease, renal disease, concurrent myopathy, and CYP2C8 polymorphisms [8].

#### 2.2.5. Neuromuscular Adverse Events

In the central nervous system, headaches, dizziness, insomnia, vertigo, tinnitus, hearing loss, and lessened seizure threshold have been described [36]. Neuropsychiatric symptoms include psychosis, delirium, personality changes, and depression [8]. In the peripheral nervous system, reversible proximal myopathy and non-painful neuropathy with bilateral proximal limb involvement, associated with hyperreflexia, and respiratory muscle involvement may occur [8,37]. The induced myopathy is dose independent, detectable by various means. Creatine phosphokinase (CPK) levels may be normal or slightly elevated, but LDH (lactate dehydrogenase), being the most sensitive in detecting muscular damage, shows increased levels [38]. Electromyography can show a neuropathic, as well as a myopathic, component, but has a low diagnostic sensitivity [39]. Muscle biopsy can exhibit mitochondrial and vacuole alterations, as well as the presence of curvilinear bodies, atrophy, muscle fiber degeneration, and necrosis [38]. Risk factors include renal disease, use of myotoxic drugs, and Caucasian status [38,39,40]. Table 2 summarizes organ-specific side effects of antimalarial therapy.

### 2.3. Monitoring

Even though side effects of antimalarial use tend to be rare and less severe than some other immunomodulatory treatments, risk of these effects can be further minimized through proper administration and monitoring. Several references have established criteria for proper use, as follows.

Ensure the daily dose does not exceed 5 mg/kg [7,41].In CKD patients with a GFR < 30 mL/min, the dosage must be adjusted to a maximum of 3 mg/kg of body weight [41].Monitor complete blood count at the beginning and during prolonged therapy [8].Surveillance of muscle strength and tendon reflexes [7].Use CPK and LDH as a screening test for myopathy and cardiomyopathy at the beginning of treatment and 3 to 6 months later [41].Monitor cQT prolongation in patients at risk [7].During the first year of treatment, use fundoscopy with visual field and spectral-domain optical coherence tomography (SDOCT) or other objective tests as needed, according to the ophthalmologist criteria, such as multifocal electroretinogram and autofluorescence imaging (in case of maculopathy) to monitor ocular toxicity [5,41].Ophthalmological control after five years of use or annually if the patient possesses risk factors [5,41].Inform patients at the beginning of therapy about possible adverse events and the importance of early recognition.Monitoring for the presence of new cardiac conduction abnormalities, biventricular and septal hypertrophy, or elevations in troponin, BNP, and CPK, can help identify patients at risk of cardiotoxicity to facilitate a diagnosis [35]. For this reason, an electrocardiogram could be performed at the start of treatment and annually.Given its safety and benefits during pregnancy and lactation, treatment should continue if indicated.

## 3. Glucocorticoids (GCs)

After inflammation is induced to handle insults to the body, anti-inflammatory homeostatic mechanisms reverse the inflammatory processes as the insulting agent is removed. The hypothalamic–pituitary–adrenal axis, through the induction of endogenous GCs (cortisol, particularly), drives this anti-inflammatory process [42].

Endogenous GCs possess broad inhibitory effects on T- and B-cell-mediated functions, as well as a potent suppressive effect on the effector functions of monocytes/macrophages, dendritic cells, and neutrophils. Hence, the endogenous GCs are essential for the immune system’s correct functioning, preventing tissue destruction and inflammatory diseases by obviating exaggerated and persistent responses to injury or infection [43,44,45]. A plethora of synthetic GCs, which mimic the potent effects of the endogenous GCs, have been developed to treat inflammatory disorders, such as asthma, allergies, sepsis, cancers, and autoimmune diseases, including rheumatoid arthritis and SLE [42,45,46]. The inhibitory effects of GCs on adaptive and innate immunologic functions, coupled with their rapid onset of action, account for their remarkable efficacy in managing the flare-ups of SLE [43,44].

The discovery of “Compound E” (hydrocortisone) in 1936 from animal adrenal gland extracts by Hench, Kendall, and Reichstein and its introduction as a clinical therapeutic agent for rheumatoid arthritis was a landmark in medical history, for which they received the Nobel Prize in Medicine and Physiology in 1950 [47,48,49]. Between 1954 and 1958, six synthetic steroids were developed for systemic anti-inflammatory therapy [49]. GCs have since become the cornerstone of SLE treatment [43]. GCs are used to treat such a wide range of inflammatory diseases that it is estimated that up to 2% of the population is receiving long-term GC therapy [50].

### 3.1. Mechanism of Action

Being highly lipophilic, GCs, after freely crossing the cell membrane, bind cytosolic GC receptors (GCRs), inducing an allosteric conformational change that results in the dissociation of the GCR from the heat shock proteins (HSPs) that chaperone the unbound GCRs to maintain their proper conformation for proper ligand binding [42,44,45,51,52]. The GC-GCR complex subsequently translocates to the nucleus in homodimeric or monomeric forms, where the immunomodulatory effects are exerted via several mechanisms [43] (see Figure 2). The first is termed transactivation. Homodimeric GC-GCR (hGC-GCR) complex binds to a specific DNA motif called glucocorticoid response element (GRE) on the promoter of glucocorticoid-responsive, anti-inflammatory genes, such as Ikβ, IL-1RII, Lipocortin-1, IL-10, and α2-macroglobulin. Binding of the GREs recruits chromatin-modifying co-factors and the transcriptional machinery to drive the anti-inflammatory genes [42,43,44,51,52,53]. In the second, termed transrepression, monomeric GC-GCR complex (mGC-GCR) binds to pro-inflammatory transcription factors, such as AP-1 and NF-kβ, inhibiting the expression of their target genes, including IL-1β, TNFα, and IL-2, cytokines, which are the major drivers of inflammation [42,43,44,51,52,53]. Additionally, prostaglandins, cytokine receptors, adhesion molecules, class II MHC molecules [43], and chemotactic proteins that play a crucial role in coordinating the inflammatory response are downregulated [54], and chemotactic proteins that play a crucial role in coordinating the inflammatory response are downregulated [55,56,57]. Another form of transrepression occurs through the GC-GCR complex binding directly to DNA sites (composite GREs), alongside AP-1 on its promoter, and hindering the expression of AP-1 target pro-inflammatory genes [44,45,52,53] (see Figure 2).

The mechanisms discussed above are collectively called genomic mechanisms because they all involve gene expression modulation. Several other mechanisms that preclude gene expression manipulations, termed non-genomic mechanisms, contribute significantly to the effects of GCs. They underlie the rapid onset of action of GCs, as they require no gene expression to impact the cell [54,58]. The best-elucidated non-genomic mechanism involves the activation of endothelial nitric oxide synthetase (eNOS) [59]. In this pathway, the GC-GCR complex activates phosphoinositide-3-kinase (PI3K) in endothelial cells, which activates Akt via phosphorylation. Phosphorylated Akt also activates eNOS via phosphorylation, resulting in nitric oxide production, which produces the physiological effects. This pathway was shown in mice to abate vascular inflammation and reduce myocardial infarct sizes following ischemia and reperfusion injury [46]. Other non-genomic mechanisms include: (1) activation of annexin I (lipocortin-1), an anti-inflammatory protein that inhibits phospholipase A_2_ (PLA_2_) and, therefore, arachidonic acid synthesis [54]; and (2) induction of the anti-inflammatory protein MAPK phosphatase 1, which inactivates all members of the MAPK protein family, including Jun N-terminal kinase and kinases 1, 2, and p38. As these MAPKs promote inflammatory pathways, their inactivation boosts the control of inflammation. Consequently, MAPK phosphatase 1 can indirectly inhibit the activity of PLA_2_ by blocking the MAPKs required for its activation and reduce the activity of lymphocytes through the p38 MAPK inhibition [42,54,58].

GCs also affect blood cell numbers; they increase the circulating neutrophil count, but decrease lymphocyte, eosinophil, basophil, and monocyte counts. The increased neutrophil count is secondary to their increased release from the bone marrow and the inhibition of their emigration [54]. The diminished circulating T cell numbers result from promoting apoptosis and migration to the bone marrow or secondary lymphoid tissues [47]. Furthermore, GCs can decrease fibroblast proliferation, fibronectin production [54], and dendritic cell maturation, survival, and migration, inhibiting their immunogenic functions, including stimulation of T cells [57].

### 3.2. Dosage

The dosage of GCs is more art than science. Albeit several organizations have published dosage guidelines, there are discrepancies between them, and standardization of GC dosage has proved challenging even till now. Consequently, physicians manage patients on a case-by-case basis, based on patient factors and their experience, guided by published recommendations [43,51,58,60]. The dosage of GC therapy determines the extent of GCR saturation. Low GC doses—i.e., prednisone doses—(up to 7.5 mg/day) are associated with up to 50% GCR saturation. Intermediate doses (>7.5–30 mg/day) achieve progressively higher saturation, with high doses (>30–100 mg/day) reaching 100% GCR saturation [43,54,58].

At very high doses (>100 mg/day), the rapid-onset non-genomic GC mechanisms are invoked [58]. Methylprednisolone (MP) and dexamethasone have non-genomic effects up to five times more potent than their genomic effects. Therefore, they act rapidly and are effectively used for intravenous (IV) pulse therapy, employed to manage severe organ and life-threatening manifestations [55,61].

SLE can be vaguely categorized as mild, moderate, or severe. The treatment of moderate to severe SLE comprises an initial phase of intensive immunosuppressive treatment called induction therapy, of which GCs (oral or IV) are central [62,63]. Induction therapy is purposed to halt active systemic inflammation and induce remission, followed by a less aggressive ‘maintenance therapy’ to consolidate remission and reduce the risk of flares [62,63]. The choice of GC dose, administration route, and duration of therapy, therefore, varies based on several factors [64]. In general, low prednisone doses are used as maintenance therapy and intermediate doses in moderate disease (fever, fatigue, weight loss, lymphadenopathy) or after MP pulses in severe SLE [43]. For instance, lymphadenopathy, arthritis, arthralgia, and myalgia can be controlled with doses of up to 20 mg/day of prednisone; however, lupus myositis cases will require higher doses of ≤60 mg/day coupled with cyclophosphamide (CYC) IV pulses [43,65,66]. In addition, Zhou et al. found that doses of ≤100 mg/day can suppress SLE-induced fever in 80.6% of patients [59]. High doses are indicated in severe manifestations [67], such as moderate cytopenia or some types of serositis. Moreover, very high doses or pulses of MP are used in life-threatening situations involving vital organs [55,56,61]. Such cases include lupus nephritis (LN), severe leukopenia or thrombocytopenia, and hemolytic or aplastic anemia. Others include gastrointestinal (autoimmune hepatitis, pancreatitis, enteritis), pulmonary (alveolar hemorrhage, shrinking lung syndrome), cardiac, and central nervous system (neuromyelitis optica, seizures, coma, peripheral neuropathy, optic neuritis, transverse myelitis) involvement [64].

In LN, the combination of medium-dose prednisone with IV pulses of MP, CYC, and HCQ is as effective as high-dose prednisone regimens, which are fraught with several adverse effects [51,68]. Ruiz-Arruza et al. compared the efficacy and safety of prednisone regimens at doses ≤30 mg/day versus >30 mg/day as initial treatment in recently diagnosed SLE patients with highly active disease without renal involvement. They found that the prednisone doses ≤30 mg/day were as effective as the higher doses for SLE treatment, but safer [69]. Accordingly, the lowest effective GC doses are increasingly preferred for treatment to reduce the risk of adverse events [64]. For this reason, the 2019 EULAR/ERA–EDTA (Joint European League Against Rheumatism and European Renal Association–European Dialysis and Transplant Association) guidelines for managing LN recommends a total intravenous MP dose of 500–2500 mg as induction therapy, then followed by oral prednisone maintenance therapy doses of 0.3–0.5 mg/kg/day for up to 4 weeks, and then reduced to ≤7.5 mg/day by 3–6 months [52,53].The current recommended doses are less than previous ones [70].

Once an SLE flare is diagnosed, the goal is to achieve remission as soon as possible and prevent new flare-ups, usually using GCs in combination with other drugs [51,63]. As the disease activity is controlled, less toxic immunosuppressive therapy is favored while the GC dose is tapered after 4–6 weeks of therapy initiation. A typical tapering starts with lowering the GC dose by 5–10 mg every 2–4 weeks until a daily dose of 20 mg, after which a reduction of 2.5–5 mg every 2–4 weeks is adopted until a maintenance dose of 2.5–10 mg/day is achieved [43]. A study showed that tapering GC doses below 5 mg have increased since 2000, probably due to a better understanding of long-term GC side effects even at low doses. The positive predictors of successful GC tapering in a cohort of SLE patients in the study were the absence of sustained skin and joint lupus activity [71].

### 3.3. Corticosteroid Resistance

In general, GC resistance is defined as the total or partial inability of cells to elicit GC responses or the absence of overt Cushing’s syndrome signs with biological hypercortisolism [45,72]. Resistance to the therapeutic effects of GCs is a considerable problem in managing inflammatory diseases [45]. In fact, up to a third of SLE patients have a partial response to GCs [51]. This underlies the marked variability of patient response to GC treatment, leading to inadequate therapy in some patients, which indicates higher doses or the addition of immunosuppressive drugs [45]. Several molecular mechanisms underlie resistance to GCs: (a) GCR loss-of-function mutations [72]; (b) decreased expression of GCRα, the GCR isoform mediating GC’s molecular effects [55]; (c) overexpression of GCRβ, a GCR isoform, functioning as a negative inhibitor of GCRα, hence, the action of GCs [43]; (d) post-translational modifications of GCR, altering its function [73]; (e) overexpression of pro-inflammatory transcription factors, such as AP-1 or NFκβ; (f) overexpression of macrophage migration inhibitory factor [74]; and (g) overexpression of P-gp (P-glycoprotein), a GC efflux pump that removes GCs from cells [43].

### 3.4. Safety

Although GCs are very potent quick-acting drugs, the concomitant damages of their use are substantial, especially with prolonged use at high doses [58,67]. Some side effects, including hyperglycemia, Cushing’s syndrome, and psychosis, are reversible. These are ameliorated by decreasing doses or therapy suspension. Others, such as cataracts, avascular osteonecrosis, and growth retardation, are irreversible [67,75]. The severity of side effects correlates with the administered doses [58]. For instance, sustained prednisone doses >7.5 mg/day were associated with increased adverse events, correlating with increased patient morbidity and permanent damage [76,77]. Moreover, a study with a cohort of 747 SLE patients linked high cumulative prednisone doses to osteoporotic fractures, coronary artery disease, and cataracts; twice-monthly high-dose prednisone to avascular necrosis and stroke; and MP IV pulses to cognitive impairment [78]. The side effects of GCs are summarized in Table 3.

#### 3.4.1. Musculoskeletal Side Effects

##### Osteoporosis

GCs have been linked to bone diseases since 1932. Due to their wide usage, GC-induced osteoporosis is the most common cause of iatrogenic osteoporosis today and, in fact, the commonest cause of osteoporosis in adults 20 to 45 years old [50,51].

Within 12 months of therapy, GCs stimulate osteoclastic activity, decreasing bone density via excessive resorption—mediated by overexpression of the receptor activator of NFκβ ligand and macrophage colony-stimulating factor—and suppressing osteoprotegerin production, which promotes osteoclastogenesis. These osteoclast-mediated effects occur first, but transiently [79].

The slower, long-lasting impact of GCs is exerted via suppression of osteoblast activity, mediated via multiple mechanisms: (a) decreased expression of Insulin-like Growth Factor-1, involved in osteoblastogenesis; (b) increased levels of the Dickkopf protein that negatively regulates the Wnt pathway involved with the differentiation, proliferation, and maturation of osteoblasts; (c) coaxing of osteoblast progenitor cells toward adipogenesis, hence, reducing osteoblast numbers; and (d) caspase 3 stimulation, which promotes apoptosis of osteoblasts and osteocytes [80,81].

Bone loss is more prominent in trabecular bone-rich areas, posing a higher risk of hip and vertebral fractures than forearm fractures. Loss of bone mass can become evident within three months of starting treatment and is correlated with high doses and longer treatment durations [82,83]. No GC dose eliminates the risk of osteoporosis. Even <2.5 mg prednisolone doses confer a higher risk of hip and vertebral fractures relative to controls [82].

##### Osteonecrosis

Osteonecrosis (ON) can result from significant reduction or interruption of the blood supply to bone, including intraluminal obstruction, vascular compression, or trauma to the vessels [84]. Several other conditions can cause ON, including SLE, sickle cell disease, pancreatitis, Gaucher’s disease, and exogenous or endogenous hypercortisolism (GC medications, Cushing’s disease) [84]. According to a meta-analysis published in 2017, the prevalence of symptomatic and asymptomatic avascular osteonecrosis (AO) in SLE patients is 9% and 29%, respectively. The most frequent site of AO is the femoral head because a terminal arterial system supplies it and it has no collateral blood supply, which renders it more susceptible to ischemia when occluded [85]. GCs induce an increase in the marrow fatty mass and fat cell sizes, resulting in intraosseous hypertension. Consequently, microvasculature occlusion by fatty emboli or impedance of sinusoidal blood flow occurs, leading to ischemia [84]. If ischemia is prolonged, necrosis progresses to sequestra formation, which results in a subchondral stress fracture, and then collapse and degenerative arthritis [84,86].

The mean daily dose and duration of GC exposure do not seem to be related to ON. Pharmacological interventions include low molecular weight heparin, lipid-lowering drugs, acetylsalicylic acid, and iloprost; however, it is not clear whether these treatments delay or reverse the disease progression. MRI is pivotal in diagnosis. The most useful MRI applications in ON diagnosis include (a) detecting early or small lesions, (b) differentiating ON from other bone diseases, and (c) predicting the likelihood of subchondral collapse. ON shows a characteristic MRI appearance for conclusive diagnosis [86].

##### Myopathy

Long-term use of GCs is associated with muscle atrophy, with decreased muscle strength mediated by two main mechanisms: reduced synthesis and increased degradation of proteins [87,88]. GC-induced myopathy primarily affects proximal muscles (e.g., the pelvic girdle muscles) [87]; yet, less frequently, distal muscles, sphincters, or facial muscles may be compromised [89]. Serum levels of CPK and aldolase are often normal, but LDH may be elevated [90]. Electromyography may present a myopathic pattern in the late stages. Muscle biopsy may show an increased number of central sarcolemma nuclei and loss of the crossed striae of type IIb muscle fibers without necrosis or inflammation, differentiating it from inflammatory myopathies [87,90].

Myopathy is uncommon in patients treated with prednisone doses of 10 mg/day, but with doses >40–60 mg/day, it can occur within the first 2 weeks of treatment [90].

Treatment suspension, physical therapy, and adequate protein intake have been shown to improve muscle strength between 3–4 weeks, although recovery may be slower [91].

##### Growth Retardation

GC-induced growth retardation is frequent in children receiving long-term GC treatment, averagely delaying skeletal maturation by 3.1 years and growth rates to only 3 cm/year [92]. GCs decrease growth hormone (GH) secretion, insulin-like growth factor I (IGF-I) bioactivity, collagen synthesis, nitrogen and mineral retention, and chondrocyte proliferation [92]. They can also induce gonadotropin, testosterone, androstenedione, and estrogen deficiency [93]. For this reason, they have been associated with delayed growth and puberty. In a cohort of 25 GC-treated children with SLE, Abdalla et al. recorded 32% growth retardation [94]. Furthermore, in the PRINTO study with juvenile SLE patients, children with early-onset disease treated with cumulative doses of GCs > 400 mg/kg had a higher risk of growth disturbances and delayed puberty [94].

Given the significant degree of growth failure in many GC-treated children, there is great interest in the potential reversal of GC-induced growth failure with GH therapy [92]. In their study, Allen et al. showed that GH therapy counterbalances the effects of GCs effectively, albeit its effectiveness was negatively correlated with GC dose. They also showed that IGF-I, IGF-binding protein-3, osteocalcin, and procollagen were appropriate markers for monitoring growth retardation and GH therapy effectiveness [92].

#### 3.4.2. Metabolic Side Effects

##### Hyperglycemia/Diabetes Mellitus (DM)

Along with chronic inflammation and obesity, GC therapy causes or exacerbates insulin resistance in non-diabetics or known diabetics, respectively [95]. The prevalence of GC-induced DM ranges between 5% and 45%; however, most studies agree that it is approximately 10–20% [96]. GC therapy increases the risk of DM by 2–3 times, the risk increasing in a dose-dependent manner [97]. In diabetics, administration of MP pulses increases the need for insulin therapy in up to 64% of patients [98]. GC-induced DM is mediated by complex mechanisms that are not well understood [99]. The effect of GCs on glucose metabolism likely results from the impairment of multiple pathways [99]. Excess GCs stimulate gluconeogenesis and glycogenolysis [100], alters insulin secretion and sensitivity in tissues [98], reduces β-cell mass [99], reduces GLUT-2 expression, inhibits GLUT4 translocation to the plasma membrane in skeletal muscle [101], and potentiates the effects of insulin-counteracting hormones, such as glucagon and epinephrine [102].

The main risk factors for developing DM include higher dosages, type of GC, longer duration of treatment, advanced age, high body mass index, family history of DM, and concurrent use of MMF (Mycophenolate Mofetil) and calcineurin inhibitors [102].

In general, GC-induced hyperglycemia improves with dose reduction; however, an individualized approach must be taken for each patient, such as lifestyle modifications and the requirement to initiate hypoglycemic drugs [102].

##### Dyslipidemia

The prevalence of dyslipidemia in SLE patients ranges from 36% at the time of diagnosis to 60% after 3 years [103], even 75% being reported in a cohort from Indonesia [104]. Sajjad et al. reported that the frequency of an altered lipid profile in SLE patients with LN of proteinuria > 1 g is increased significantly [105]. Dyslipidemia in SLE patients is associated with cardiovascular events and aggravation of kidney and central nervous system damage [106]. Many factors influence dyslipidemia development in SLE patients, such as autoantibodies, cytokines, and GC and cyclosporine A treatment [98,104,107]. The effects of GC on lipid metabolism are not well understood. However, it is known that cortisol activates lipolysis; increases triglycerides (TG) hydrolysis in adipocytes, free fatty acid levels [98,102], lipoprotein lipase and adipokine activity, and insulin resistance; and inhibits beta-oxidation of lipids [103]. Additionally, GCs induce changes in lipoprotein metabolism, stimulating the production of very-low-density lipoproteins (VLDL) and HDL and inhibiting the uptake of LDL [108]. As such, the lipid profile should be monitored in all SLE patients. Lipid-lowering drugs, mainly statins, should be administered when necessary to reduce coronary heart disease, cerebrovascular disease, kidney disease, and mortality [105,109].

While low doses equivalent to prednisolone < 10 mg/day do not significantly affect the lipid profile [109], doses ≥30 mg/dL are associated with high levels of total cholesterol (TC) and TG [104], but have a weak influence on LDL and HDL [108].

##### Weight Gain and Lipodystrophy

During systemic GC therapy, weight gain and morphological changes secondary to adipose tissue accumulation are frequently observed. This fatty tissue accumulation is often seen in the face (“full moon face”), dorsal-cervical area (“buffalo hump”), supraclavicular, and abdominal regions. However, there is a decrease in the subcutaneous fat of the extremities [110]. In a cohort of 236 SLE adolescent patients, 90% had a normal BMI at the beginning of GC therapy, but by the end, approximately 20% had a BMI > 25, and 10% were obese. Overall, 60% gained less than 10 kg, 25% gained 10–20 kg, and 15% gained more than 20 kg after treatment [111].

Cushingoid features can develop within the first two months of therapy. As many as 15–40% of patients may present with “moon face” after just 8–12 weeks of prednisone treatment (doses of 10–30 mg/day) [110].

The risk of these complications appears to depend on both the dose and the duration of treatment. In a cohort of 88 patients put on long-term systemic GC treatment, incidence rates increased over time. The risk of lipodystrophy was higher in patients who were women, were under 50 years of age, had a high BMI at the beginning of treatment, or had high caloric intakes [110].

The pathophysiologic mechanisms are multifactorial. They include the mechanisms of dyslipidemia and hyperglycemia described above; induced hormonal changes in growth hormone, testosterone, estrogens, catecholamines, and cytokines [112]; as well as stimulation of orexigenic pathways in the hypothalamus [98].

#### 3.4.3. Cardiovascular Side Effects

##### Arterial Hypertension

Overall, about 20–30% of patients undergoing long-term GC therapy suffer GC-induced hypertension [100,113], and the incidence rates increase with higher cumulative doses [114]. L Fardet suggests that there may be two forms of arterial hypertension associated with GC therapy—an early-onset type (within days to weeks of treatment) in patients without risk factors and a late-onset type in patients with drug-induced lipodystrophy and weight gain [98]. Proposed mechanisms include an increased transcription of genes (*sgk-1*, *α-ENaC*, and *GILZ*) responsible for sodium reabsorption in the renal tubules and a decreased expression of the endothelial nitric oxide synthase. Other mechanisms include increased oxidative stress [115], increased expression of type I angiotensin II receptors, and stimulation of Na^+^ and Ca^2+^ entry into endothelial cells [116].

##### Cardiovascular Risk

Patients with SLE treated with GCs at a dose greater than 10 mg/day or those with a cumulative dose equivalent to more than 10 mg/day for more than 10 years have significantly higher rates of cardiovascular events [117]. The use of oral GC is associated with heart failure [118], explained by sodium retention, increased extracellular fluid, stimulation of cardiac remodeling and fibrosis, increased myocardial oxidative stress, and coronary vascular inflammation mediated by mineralocorticoid receptors [119].

#### 3.4.4. Adrenal Insufficiency

Exogenous GC administration generates a negative regulation of the hypothalamic–pituitary–adrenal (HPA) axis [106]. Minutes after GC administration, the increase in cortisol levels inhibits the release of ACTH and CRH. Later on (2–20 h), the transcription of pro-opiomelanocortin transcription factors (POMC) is inhibited, which leads to decreased ACTH synthesis, consequently reducing endogenous cortisol secretion by the adrenal gland [120].

The mean prevalence of adrenal insufficiency associated with GC use is 37% [106]. It occurs in up to one third of patients treated with 5 mg prednisolone/day, and the prevalence increases in patients who receive topical, intramuscular, or intra-articular GCs concurrently [121]. It most often occurs when therapy is discontinued abruptly or in acutely stressful situations. Physicians, therefore, wean patients off GCs by tapering the doses. After stopping treatment, HPA axis suppression may occur, with or without clinical manifestations such as asthenia, adynamia, nausea, abdominal pain, headache, or dizziness [122].

There is a great variety of information regarding the axis recovery time after therapy discontinuation. The earliest recovery time is 4 weeks [120], but axis suppression can persist for 24 months [123].

#### 3.4.5. Skin Disorders

GCs cause a reduction in the mitotic activity of keratinocytes; reduce the size of fibroblasts, cause thinning of the dermis; and increase the fragility of the skin. Additionally, they cause a decline in monocyte and macrophage count, diminish phagocytosis, and delay re-epithelialization and fibroblast response [89].

The reported dermatological conditions include rosacea, erythema, telangiectasias, acneiform eruptions, purpura, pruritus, atrophy, hirsutism, stretch marks, decreased healing, and dermatitis [123]. Atrophy and ecchymoses are often reversible with GC therapy suspension, but stretch marks persist [89]. Purpura generally affects sun-exposed areas, such as the neck, back of the hands and forearms, face, and lower legs [79].

#### 3.4.6. Neuropsychiatric Disorders

GCs can induce neuropsychiatric manifestations, such as depression, hypomania, psychosis, bipolar disorders, delirium, panic attacks, agoraphobia, obsessive–compulsive disorder, anxiety, insomnia, restlessness, catatonia, and cognitive impairment [124]. Symptoms can become severe in 5% of the treated patients [125]. They can appear within the first 6 weeks of treatment; in fact, some cohorts have reported 86% of patients presenting symptoms in the first week [126].

Female gender, SLE, and high doses may be risk factors for the development of symptoms [125]. Dose reduction or gradual suspension of GC is the mainstay of management; up to 90% of patients improve within the first 6 weeks of recess [127].

Some of the proposed pathophysiological mechanisms are the downregulation of GCR, the induction of neuronal oxidative stress, decreased serotonin levels, increased dopamine, and decreased sex steroid production [127].

#### 3.4.7. Ophthalmic Alterations

The administration of systemic GC can lead to the formation of cataracts, glaucoma, myopia, exophthalmos, papilledema, chorioretinopathy, and subconjunctival hemorrhages [89]. GC-induced glaucoma is a form of open-angle glaucoma generated by morphological alterations of the trabecular meshwork, an increase in extracellular matrix proteins, and a decrease in vasodilator prostaglandins, which results in a diminished net output of aqueous humor. Risk factors include myopia, a history of penetrating keratoplasty or refractive surgery, patients under 10 years of age or the elderly, a history of diabetes mellitus, and endogenous hypercortisolism [128,129].

The frequency of oral GC-induced cataracts varies between 11% and 15% [129]. The risk is dependent on the dose and duration of therapy, and accrued damage is irreversible, even with treatment withdrawal. The mechanisms involved include enzymatic and cellular modifications, oxidative stress, protein alteration, and the action of various growth factors [128].

Central serous chorioretinopathy is a disorder characterized by neurosensory retinal detachment associated with detachment of the retinal pigment epithelium that can occur in patients with SLE [130], GC therapy being one of the main risk factors [131].

### 3.5. Safety in Pregnancy

GCs are one of the main treatments for lupus flares during gestation, as many other drugs are incompatible with pregnancy. They retain their potent anti-inflammatory effects without any significant teratogenicity [18]. The GC of choice depends on whether the goal is to treat the mother or the fetus [132,133]. Non-fluorinated GCs, such as prednisone, prednisolone, and MP, are the suitable GCs for treating the mother as they are inactivated by placental hydroxylases. Fluorinated GCs, such as betamethasone and dexamethasone, are less metabolized by the placenta. Hence, they are preferred if the fetus is the target of the treatment, especially in patients at risk of preterm birth, between 24 and 34 weeks of gestation, where induction of fetal lung maturation would be required [132,133].

The use of high GC doses during pregnancy is associated with an increased risk of complications, including infections, gestational diabetes, hypertension, preeclampsia, premature rupture of membranes, and the risk of cleft palate. For this reason, it is recommended to use the lowest possible dose for the shortest time, ideally a dose <20 mg/day. Hydrocortisone administration is recommended at delivery in patients on long-term GC therapy to reduce the risk of adrenal insufficiency [133].

During lactation, moderate doses are recommended, and at least a 4 h gap is to elapse after drug intake before breastfeeding [132].

### 3.6. Monitoring

As GCs have devastating effects, patients on GC treatment must be carefully monitored as follows:Determine anthropometric measurements, blood pressure, metabolic profile (glucose, glycosylated hemoglobin, LDL, HDL, TC, TG, apoB), and densitometry at the beginning of treatment [79].Guidelines for a healthier lifestyle, such as diet, regular physical activity, avoiding smoking, and reducing alcohol consumption [79].Monitor blood glucose at least 48 h after the start of therapy, then every 3–6 months during the first year, and then annually [134].Patients receiving prednisone doses >7.5 mg/day for more than 3 months should be prescribed calcium and vitamin D supplements [134].Use FRAX scores to evaluate the risk of fractures at 10 years [79].Determine anthropometric measurements in each consultation [134].Perform bone densitometry at the start of therapy and annually if there is a decrease in bone mineral density or biannually if it remains stable [134].X-ray of the lateral spine in patients ≥65 years for early detection of vertebral fractures [134].Determine if the patient requires bisphosphonate therapy according to risk factors and bone mineral density [134].Monitor lipid profile after 1 month of treatment, then every 6 to 12 months.Assess cardiovascular risk periodically.Perform bone densitometry and lateral column radiography in children receiving ≥3 months of GC therapy and repeat annually.Monitor the growth rates of children and adolescents, and refer to endocrinology, if necessary, to ascertain if growth hormone therapy is needed [134].Request an annual ophthalmological evaluation or earlier if there are risk factors or symptoms [134].Monitor blood pressure, signs of fluid overload, and heart failure at each visit [134].Watch for signs/symptoms of adverse reactions during therapy.Patients treated with a GC concurrently with a nonsteroidal anti-inflammatory drug should receive gastroprotection with proton pump inhibitors or misoprostol. Alternatively, they could switch to a selective cyclooxygenase-2 inhibitor (taking into account increased cardiovascular risk) [134].In patients requiring more than 10 mg prednisone/day, other less toxic immunosuppressants should be combined with GCs to accomplish quick tapering of prednisone and ultimately reduce GC-associated organ damage [58,62,135]. The immunosuppressants play an essential role in managing severe SLE manifestations, minimizing the risk of organ damage, reducing the cumulative dose of GCs, and preventing new flares of the disease [135]. Among the agents used are CYC (Cyclophosphamide), AZA (azathioprine), MMF (Mycophenolate Mofetil), Tacrolimus (TAC), and Methotrexate (MTX) [135].

## 4. Cyclophosphamide (CYC)

Developed by the German chemist Norbert Brock in 1958, CYC is an alkylating immunosuppressant derived from nitrogen mustard [136]. CYC was first used to treat SLE in the 1970s when Donadio et al. demonstrated that patients receiving prednisolone with oral CYC were more likely to have better renal preservation than GC monotherapy [137]. As with all other immunosuppressants, CYC is used in combination with GCs as oral or IV formulations [63,99].

### 4.1. Mechanism of Action

CYC is a prodrug predominantly (70–80%) hydroxylated by hepatic cytochrome P450 enzymes (CYP2B6, CYP2C9, CYP2C19, CYP3A4, CYP3A5, or CYP2J2) to 4-hydroxycyclophosphamide and its tautomer aldophosphamide. These metabolites enter target cells by simple diffusion [138], where 4-hydroxycyclophosphamide is inactivated. Aldophosphamide undergoes spontaneous non-enzymatic β-elimination, generating the active metabolite phosphoramide mustard and acrolein as a by-product. The former mediates the pharmacological effect of CYC, but the latter is urotoxic [136].

Phosphoramide mustard is a potent DNA alkylating agent that readily forms irreversible covalent bonds with N7 of guanine, leading to interstrand cross-links [139]. It can also bind other purine and pyrimidine atoms, blocking DNA replication and leading to apoptosis [140]. These actions exert a cytotoxic effect on actively proliferating cells, including mainly the less mature B lymphocytes, reducing antibody production by these lymphocytes [140]. Additionally, CYC dwindles the number of circulating effector T cells CD8+ CD44+ CD62L− and CD8+ CD44+ CD62L− [140].

### 4.2. Efficacy

CYC is a potent, but aggressive, drug; hence, it is only indicated for severe organ-threatening disease, especially neuropsychiatric SLE (NPSLE), cardiopulmonary, and renal compromise, where the toxicity-to-benefit ratio is justifiable [135]. It may also be indicated as rescue therapy in refractory manifestations of non-major organs [135].

A retrospective study that included 50 cases of CYC-treated patients with neuropsychiatric manifestations, such as psychosis, polyneuropathy, cerebrovascular disease, seizures, or cranial neuropathy, observed a partial or complete response in 84% of cases [141]. In a systematic review published by Cochrane in 2013 that compared CYC versus MP as NPSLE treatment, 94.7% of CYC-treated patients responded to treatment. Moreover, CYC was associated with a reduction in prednisone dose requirements [142].

GC monotherapy or in combination with CYC, MMF, or AZA is recommended in interstitial lung disease and constricted lung syndrome associated with SLE. However, this recommendation is mainly based on expert opinion as there is only tenuous evidence [142,143].

In the treatment of LN classes III, IV, and V, the combination of high-dose GC with low-dose CYC (500 mg IV bolus administered every 2 weeks for 3 months) or oral MMF (2 to 3 g/day for 6 months) is suggested [17]. High-dose CYC should be reserved for severe cases, such as rapidly progressive glomerulonephritis, where serum creatinine is >3 mg/dL with crescents or fibrinoid necrosis, or in those irresponsive to treatment [144]. However, in patients with a creatinine clearance of less than 30 mL/min, the dose should be reduced by about 30% [135]. The caution taken with high-dose CYC is due to its association with cervical neoplasms and ovarian failure [137] without superior efficacy than low doses, as evidenced by the multicenter prospective clinical trial (Euro-Lupus Nephritis Trial, ELNT). The ELNT trial compared high-dose and low-dose CYC IV regimens in patients with LN, followed by maintenance therapy with AZA. Treatment failure occurred in 16% and 20% of the low-dose and high-dose groups, respectively. In addition, renal remission was achieved in 71% and 54% of the low-dose and high-dose groups, respectively. While the efficacies were similar, episodes of severe infection occurred more frequently in the high-dose group [145].

CYC may be taken orally or intravenously, but IV pulse is preferred due to its superior efficacy-to-toxicity ratio [2]. Daily oral CYC as induction therapy may be more effective than intravenous pulses; however, its greater ovarian toxicity makes it justified only in high-risk or refractory LN [2]. Moreover, some studies suggest that CYC may have efficacy differences between different races of people [135]. For example, Dooley et al. [146] found a poorer renal survival in African Americans during the initial period of monthly IV CYC administration, with many of them rapidly progressing to renal failure. Further disparity was observed in long-term follow-up studies, with renal survival after 5 years at 94.5% for Caucasians and 57% for African Americans [146].

### 4.3. Safety

Notwithstanding its significant toxicity, CYC remains a mainstay of treatment for severe SLE. Its clinical effects (therapeutic or toxic) vary, depending on the dose, route of administration, duration of administration, and cumulative dose [135]. In the past two decades, minimizing the use of CYC for even the most severe SLE manifestations (particularly in LN) has assumed utmost importance. The main approaches for achieving this goal include: (1) using sequential therapy with CYC for induction of remission, followed by maintenance therapy with MMF or AZA; (2) shortening the period of induction with CYC; and (3) substituting MMF for CYC as induction therapy in LN [135]. The main side effects of CYC are compiled in Table 4.

#### 4.3.1. Gastrointestinal Events

The most frequent CYC adverse effects are gastrointestinal-related, such as nausea, GI dysmotility, and emesis shortly after administration, especially with the dose ranges used for SLE treatment [147]. CYC is strongly emetic; hence, the American Society of Clinical Oncology recommends antiemetics, such as Ondansetron, a potent 5-hydroxytryptamine receptor antagonist, to alleviate emesis [147]. Furthermore, serious hepatotoxicity may occasionally occur with the doses used for autoimmune diseases [135].

#### 4.3.2. Gonadal Insufficiency

Gonadal insufficiency is a significant side effect of CYC in both men and women. Amenorrhea may occur after treatment in 25–77% of treated women. The risk of ovarian failure is higher among older women and lower in patients receiving low doses. Ovarian failure results from a reduction in the number of granulosa cells, follicle sizes, maturation of oocytes, and levels of estradiol and progesterone [148]. In men, prolonged or permanent oligospermia or azoospermia has been observed [149]. Due to the risk of gonadal insufficiency, therapeutic alternatives, such as biologics, are suggested in patients of childbearing age. However, if CYC must be used, gonadotropin-releasing hormone analogs (GNRHa) should be combined with treatment [2]. GnRHa can exert direct protective effects on the ovaries through peripheral GnRH receptors and significantly reduce the risk of ovarian failure in young women with severe SLE [149]. CYC treatment is strongly associated with azoospermia; therefore, sperm banking before therapy should be considered. Additionally, testosterone supplementation during treatment helps preserve testicular functions and fertility [150,151].

#### 4.3.3. Urotoxicity

Acrolein is an extremely reactive CYC metabolite [152]. In the urogenital epithelium, it promotes the intracellular production of reactive oxygen species (ROS) and nitric oxide (RNOS), which cause oxidative stress, lipid peroxidation, and mitochondrial dysfunction. The ROS and RNOS also promote protein-DNA adducts formation that causes inflammation, necrosis of the bladder mucosa, and gross hematuria [153]. Hemorrhagic cystitis (HC) can occur in 4–36% of CYC-treated patients with autoimmune diseases [151], with the cumulative CYC dose being the most important predictor for its presentation [154]. HC is considered a premalignant lesion that can eventually progress to transitional cell carcinoma of the urinary tract or fibrosis of the bladder, requiring the definitive interruption of treatment [155]. Patients should be advised to consume copious fluids or be given intravenous fluids with CYC administration to dilute the toxic metabolites in the urine to avoid HC. Patients receiving pulsed cyclophosphamide may simultaneously receive oral or IV sodium 2-mercaptoethanesulfonate (MESNA) at 20–40% of CYC dose, which will slow down the metabolism of 4-hydroxymetabolites and help detoxify acrolein in urine [156]. CYC increases the risk of bladder carcinoma and cervical intraepithelial neoplasms [147]. Daily CYC intake is associated with a heightened risk of bladder carcinoma and is dependent on the dose and duration of exposure. IV CYC regimens have lower total doses than prolonged daily oral regimens, and their associated incidence of bladder cancer may be lower because it is typically coupled with MESNA [154,157].

Additionally, development of non-urinary tract cancers in CYC-treated patients with rheumatic diseases, such as rheumatoid arthritis, is not uncommon. Neoplastic complications, including skin and hematologic malignancies and cervical atypia, are probable, even in patient treated with cumulative doses less than 10 g [135].

#### 4.3.4. Infections

CYC can induce neutropenia, lymphopenia, thrombocytopenia, and anemia [158]. After CYC IV therapy, the lymphocyte count nadir occurs within approximately 7–10 days and granulocytes between 10 and 14 days. These counts typically recuperate after 21 to 28 days [135]; however, severe hematologic toxicity may occur in patients with polymorphisms of CYP2B6, GSTA1, and GSTP1 [159].

The frequency of infections—bacterial, herpes zoster, fungi, and some opportunistic infections (e.g., *P. carinii),* being the most reported—is about 37% [158,160]. The prevalence of infection is similar between IV CYC (39%) and oral (40%), the risk factors including leukocyte nadir ≤3000 cells, sequential CYC regimens, and combination with high-dose GC [161].

#### 4.3.5. Pulmonary Toxicity

Adverse events affecting the pulmonary system occur in less than 1% of treated patients. They manifest as early-onset interstitial pneumonitis (within six months of starting treatment) or as late fibrosis [162]. Acute interstitial pneumonitis may mimic new lupus pulmonary manifestations in a patient with active disease, making it difficult to diagnose, and late-onset fibrosis may insidiously develop after months to years of CYC therapy [135].

#### 4.3.6. Cardiac Toxicity

Oxidative stress and activation of the inflammatory pathway via NFκβ, with the simultaneous release of pro-inflammatory cytokines (IL-2, IL-10, IL-6, and TNF-α) associated with acrolein, induces hypertrophy, myocardial fibrosis, and arrhythmogenesis [163]. The clinical manifestations include tachyarrhythmias, hypotension, heart failure, myocarditis, and perimyocarditis. Albeit cardiac toxicity is rare with CYC treatment regimens in SLE, it is more frequent in oncology regimens [158].

### 4.4. Safety in Pregnancy

Exposure to cyclophosphamide during the first trimester of pregnancy can lead to spontaneous abortion or congenital malformations, including growth restriction, ear and craniofacial abnormalities, absence of fingers, hypoplastic limbs, exophthalmos, cleft palate, and skeletal abnormalities [164].

### 4.5. Monitoring

Due to the potentially severe toxicity of cyclophosphamide, the following monitoring regimens are of the highest importance.

Rule out pregnancy in women of childbearing age before starting therapy [165].Advise women of childbearing potential to use effective contraception during treatment with cyclophosphamide and for up to one year after the last dose [165].Recommend male patients with female partners of reproductive potential to use effective contraception during CYC treatment and for four months after the last dose.Inform patients about the possible risks of infertility with therapy [165].Perform a baseline blood count weekly for the first four weeks, every two weeks until the second month, and monthly thereafter. Do not start treatment in patients with an absolute neutrophil count of <1500/mm^3^ and platelets of <50,000/mm^3^ [166].Correct or exclude any type of urinary obstruction because this may increase the risk of urotoxicity [166].Perform urinalysis to evaluate the presence of hematuria, proteinuria, or bacterial infections. This test is initially recommended weekly for the first four weeks, then twice weekly until the second month, and monthly thereafter.Surveillance for signs/symptoms of infection.Monitor for signs and symptoms of cardiotoxicity or pulmonary toxicity [167].

## 5. Azathioprine (AZA)

AZA is one of the oldest immunosuppressive agents in use, having been used for several decades [135,168]. It is a purine analog developed from the anti-cancer agent 6-mercaptopurine (6-MP), initially purported to be a long-lived pro-drug version of 6-MP for better chemotherapy [168,169]. It was soon found to possess a better therapeutic index and effectively induced remission in childhood acute leukemia [168]. Later, it was shown to have immunosuppressive properties, such as reducing antibody production, prolonging allograft survival in transplant patients, reducing the severity of experimental lupus nephritis, and showing efficacy in treating rheumatologic diseases [168,169]. AZA is currently a valuable immunosuppressant for managing multiple SLE manifestations and a myriad of hematologic malignancies, rheumatologic disorders, and solid organ transplantation [168,170,171]. AZA is the only drug in its class currently in wide use for SLE management [171].

### 5.1. Mechanism of Action

Although the immunomodulatory mechanism of AZA is not well elucidated, its generally accepted to be mediated by DNA synthesis inhibition [171]. After its absorption, AZA is first non-enzymatically reduced by glutathione to 6-MP and then enzymatically converted to 6-thioinosinic acid (6-TIA), 6-thiouric acid (6-TUA), 6-methyl-MP (6-MMP), and 6-thioguanine (6-TG), which are collectively called thioguanine nucleotides (TGNs) [135,168,172]. The TGNs (6-TG and 6-TIA) block the de novo purine synthesis pathway and, ultimately, DNA synthesis by incorporation. Blocking the de novo purine synthesis is thought to underlie AZA’s relative specificity to lymphocytes as they lack a salvage purine synthesis pathway; however, the DNA synthesis blockade alone does not sufficiently explain all the clinical findings of AZA-induced immunosuppression [168]. For instance, AZA reduces the levels of T cells, B cells, and natural killer cells, inhibiting both cellular and humoral immunity, as well as suppressing autoantibody formation and prostaglandin synthesis [173].

Other mechanisms contributing to AZA-induced immunosuppression, such as the following. (1) Direct apoptosis of T cells and inhibition of cell migration: in vitro studies showed that AZA and its metabolite, 6-TG triphosphate, interact with and block RAC1, a GTPase functioning in T-cell activation pathways, survival, migration, and adhesion. By blocking RAC1, all RAC1 target genes crucial for inflammation, T-cell activation, and survival, such as NF-κβ, mitogen-activated protein kinase (MAPK), and bcl-X_L_, a protein complex with antiapoptotic properties, are suppressed [174]. Therefore, AZA surges T-cell susceptibility to apoptosis [174]. (2) Decreased synthesis of inducible nitric oxide synthase (iNOS): another in vitro study has shown that AZA can block RAC1 action in macrophages, a function necessary for iNOS expression. This blockade reduces iNOS mRNA levels, which is associated with decreased expression of IRF-1 (interferon regulatory factor 1) and IFN-beta (beta-interferon) mRNA. Hence, the inhibition of iNOS might contribute to the anti-inflammatory properties of AZA [174,175].

### 5.2. Efficacy

Despite several decades of clinical use, AZA has not been established as a first-line drug in severe SLE treatment [135]. In LN, it is most effectively used as a maintenance or steroid-sparing agent (2–3 mg/kg/day) employed after induction of remission with more potent and faster-acting agents, such as CYC or MMF [171]. Following the MAINTAIN trial, MMF usurped AZA as the preferred treatment in LN. However, AZA has found its niche in predominantly female patient populations of child-bearing age, as it is one of the few immunosuppressants deemed safe during pregnancy [176]. Indeed, it is considered the first-choice drug in pregnant patients [171,176].

The remission-maintaining benefits of AZA are not restricted to LN SLE manifestations alone. It is generally prescribed in SLE cases without renal involvement, where recurrent flares occur, due to its ability to reduce the frequency of flares [171,176]. It is reported to be effective in managing severe cutaneous SLE, autoimmune hepatitis, inflammatory bowel disease (IBD), and organ transplantation [135,170,173]. AZA is also efficacious as maintenance therapy in neuropsychiatric SLE and rheumatoid arthritis; however, it is not well tolerated with arthritis [177].

### 5.3. Safety

#### 5.3.1. Genetic Predispositions

In hematopoietic cells, the primary enzyme that metabolizes AZA to its final active metabolites (TGNs) is thiopurine methyltransferase (TPMT). It has been shown that TGN accumulation in cells, which is inversely related to TPMT activity, is a significant determinant of AZA’s toxicity and efficacy [172,178]. Interestingly, population studies show that 1 in 300 patients lack TPMT activity, and 10% have partial activity. These patients have genetic polymorphisms of TPMT that make for poor AZA metabolism; thus, they have an increased risk of toxicity or failed treatment [62,172,178]. Hence, testing for TPMT is recommended to help predict efficacy and drug-induced toxicity in AZA-treated patients with such polymorphisms [62]. However, pre-treatment TPMT genotyping or phenotyping is not widely implemented in rheumatology because it lacks consistency and may not identify many patients who eventually develop myelotoxicity [170,178]. The side effects of AZA are summarized in Table 5.

#### 5.3.2. Hematological Effects

Myelosuppression is a significant complication of AZA treatment. Leucopenia and thrombocytopenia complications occur in up to 27% and 5% of AZA-treated patients, respectively. The risk of these myelosuppressive side effects is greatest in patients with low TPMT activity [170], although it also occurs in patients with normal TPMT activity [135,178]. Low or no TPMT function leads to the accrual of higher intracellular concentrations of TGNs, increasing the risk of severe myelosuppression [178,179]. The MAINTAIN study showed that AZA induces hematological cytopenias more frequently than other drugs, such as MMF [176], eliciting subsequent complications, including sepsis, severe anemia, and bleeding [180]. Yet, mild symptoms are usually reversible with treatment withdrawal and are dose-dependent [170,178]. Concurrent usage with allopurinol, febuxostat, xanthine oxidase inhibitors, or ACE inhibitors augments the risk of myelosuppression by altering the balance between active and inactive metabolite levels [178,181]. Some authors recommend a switch to a different medication or tapering the doses in severe cases [170,180].

Reports of AZA causing neoplasms remain controversial, as the many studies designed to answer this conundrum have generally been underpowered and only yielded conflicting results. The preponderance of reported cancers in the literature is non-Hodgkin’s lymphoma. Many of these lymphomas have been associated with Epstein–Barr virus (EBV) and immunosuppression, and some lymphomas resolve after treatment cessation [178].

#### 5.3.3. Increased Risk of Infections

Like all immunosuppressive regimens, AZA increases the risk of infections. An extensive comparative study of SLE patients taking immunosuppressants found no significant difference between the infection rates of AZA (17.8%) and MMF (17.4%) [182]. Due to its cytotoxic effects on lymphocytes, AZA has a propensity for causing viral infections, including EBV, cytomegalovirus (CMV), or varicella-zoster virus (VZV) [178]. However, some studies suggest that bacterial infections are more common [182]. In contrast, studies show that MMF-AZA combination therapy gives a very low risk of severe infections compared to CYC, GCs, or either of them alone, being matched only by TAC [183].

As there is potentially an enhanced risk of vaccine-preventable infections in AZA-treated patients with rheumatological diseases, such as SLE and IBD, a vaccination strategy is essential. The European Crohn’s and Colitis Organization (ECCO) guidelines espouse an intensive screening and vaccination program for infections, including VZV, pneumococcal, hepatitis B, HPV, and influenza, at the time of diagnosis for such patients as a preventive strategy. However, they advise against the use of live vaccines [184].

#### 5.3.4. Gastrointestinal Effects

Gastrointestinal (GI) intolerance is the most prevalent side effect with AZA therapy, accounting for about 10% of treatment discontinuation [180]. AZA instigates several well-documented GI symptoms, including anorexia, nausea, vomiting, and diarrhea, occasionally severe enough to warrant therapy cessation [62,185].

AZA-induced pancreatitis rarely occurs in SLE patients; however, it affects 2–7% of IBD patients. It appears in a dose-dependent but unpredictable manner. Risk factors associated with onset include GC treatment and cigarette smoking; however, no predictive clinical tests are available to identify at-risk patients [185,186]. Yet, a recent retrospective study involving 373 AZA-exposed IBD patients by Wilson et al. has revealed that predictability is possible after all [185]. They showed in their work that single nucleotide polymorphisms in the class II HLA gene region at rs2647087 mapped to the *HLA-DQA1*02:01-HLA-DRB1*07:01* haplotype was a useful marker and predictor of AZA-induced pancreatitis. Accordingly, they proposed that a genotype-guided treatment algorithm be implemented to obviate adverse reactions [185].

Another significant AZA-induced complication is hepatotoxicity [187]. It has long been associated with AZA therapy and demonstrated as a dose-dependent and reversible (when the inciting agent is removed) phenomenon [178]. Generally, AZA-mediated liver toxicity arises within 12 months of therapy initiation. The overall incidence ranges from <1–10%, and about 90% of cases occur in males [187]. Several markers on the liver function test panel, including the liver enzymes, are usually elevated, which can mimic cholestatic hepatitis [187].

### 5.4. Safety in Pregnancy and Lactation

Notwithstanding the United States food and drugs board’s classification of AZA as a class D agent (potentially harmful to the fetus, hence should be prescribed during pregnancy only after careful evaluation of risk versus benefit), it is considered the first-choice immunosuppressant during pregnancy [135,178,188]. The teratogenicity of AZA has been established in mice and rabbits, but human fetuses lack the requisite enzymes to convert the pro-drug (AZA) into active metabolites, hence, deemed protected from the disfiguring effects [178,189]. Some studies have corroborated this claim; however, some studies show that some mild complications, such as developmental delays, pancytopenia, increased risk of premature birth, and mild malformations, may occur [178,189,190].

AZA and its metabolites are detectable in breast milk; therefore, breastfeeding is ill-advised. However, some later studies show that AZA levels in breast milk are diminished significantly within four hours of drug intake; hence AZA treatment is compatible with breastfeeding [178,191,192].

### 5.5. Considerations in Renal Insufficiency

AZA is principally eliminated through the kidneys, and it has a short elimination half-life between 60–120 min after its conversion to 6-MP [193]. Although it is very effective as maintenance therapy (comparable to MMF) in treating LN [176], patients with KDIGO 3 chronic kidney disease have a higher risk of developing adverse reactions [194]. Where adverse effects occur, it is recommended to cut the dose by 75% for patients with estimated glomerular filtration rates of <50 mL/min/m^2^ calculated by the Cockcroft-Gault equation. For patients with renal replacement therapy, such as hemodialysis, 50% of the dose should be administered before the procedure and supplemented with 0.25 mg/kg afterwards [194,195].

### 5.6. Monitoring

To prevent or minimize potential side effects of azathioprine administration, the following monitoring mechanisms are useful.

Consider genotyping or phenotyping patients for TPMT deficiency and genotyping for NUDT15 deficiency in patients who develop severe myelosuppression [195].Monitor hemogram, including platelet counts weekly during the first month, twice monthly for the second and third months of treatment, then monthly or more frequently if dosage alterations or other therapy changes are necessary [196].Liver function tests should be monitored periodically during therapy for early detection of hepatotoxicity https://www-micromedexsolutions-com.roseman.idm.oclc.org/micromedex2/librarian/CS/7F44AE/ND_PR/evidencexpert/ND_P/evidencexpert/DUPLICATIONSHIELDSYNC/217C6E/ND_PG/evidencexpert/ND_B/evidencexpert/ND_AppProduct/evidencexpert/ND_T/evidencexpert/PFActionId/evidencexpert.DoIntegratedSearch?SearchTerm=Azathioprine&fromInterSaltBase=true&UserMdxSearchTerm=%24userMdxSearchTerm&false=null&=null-cite97_de, every two weeks for the first four weeks and monthly thereafter [196].Surveillance for signs/symptoms of infection.

## 6. Mycophenolate

MMF, synthesized around 1990, is an ester prodrug derivative of mycophenolic acid (MPA), an immunosuppressant used initially to avert rejection in kidney, heart, and liver transplantation, later employed to treat rheumatic diseases. The first MMF clinical trial in SLE was undertaken around the year 2000, and it is now considered a standard treatment for LN [197]. There are also clinical trials demonstrating its efficacy in treating non-renal SLE manifestations, such as arthritis, skin, or hematological involvement [198]. MMF is usually administered at a fixed oral dosage, and side effect monitoring is not routinely performed. However, MMF administration is sometimes associated with tolerability problems due to gastrointestinal adverse effects, such as vomiting, diarrhea, abdominal pain, and gastritis [199]. Hence, another MPA derivative, mycophenolate sodium (MPS), has been developed to tackle the side effects of MMF. The enteric-coated formulation of MPS (EC-MPS) releases MPA in the small intestine instead of the stomach, therefore, reducing MPA-related upper gastrointestinal adverse events [2].

### 6.1. Mechanism of Action

MMF, being a prodrug, is metabolized to its active form, MPA, which inhibits inosine 5-monophosphate dehydrogenase (IMPDH), an essential enzyme for guanine nucleotide synthesis. Thus, MMF causes the dwindling of B cell, T cell, and fibroblast numbers by inhibiting guanine synthesis. In addition, MMF has antifibrotic effects by reducing serum concentrations of transforming growth factor β (TGFβ), fibronectin synthesis, and proliferation of mesangial cells involved in the pathogenesis of renal fibrosis [173].

MPA has also been shown to inhibit the expression and function of cell adhesion molecules, thereby hindering the recruitment of lymphocytes and monocytes to sites of inflammation. It also induces apoptosis of activated T lymphocytes and suppresses nitric oxide production by reducing inducible nitric oxide synthase activity [200].

### 6.2. Efficacy

MMF or CYC combined with high dose GC are used as the induction and maintenance regimens for LN classes III, IV, and V. The ALMS study (Aspreva Lupus Management Study), involving 370 patients with LN classes III-V, compared MMF (3 g/day) with CYC (0.5–1.0 g/m^2^ in monthly pulses) as induction therapy and showed that MMF and CYC had similar efficacies at 6 months and after 3.5 years. No significant differences were detected between the groups concerning the rates of serious adverse events or infections [201]. However, race, ethnicity, and geographic region were shown to affect response to treatment—more black and Hispanic patients responded to MMF than to CYC [201]. Rathi et al. compared MMF (1.5–3 g/day for 24 weeks) with low CYC doses (6 infusions of 500 mg every 15 days) as induction therapy in LN. All patients also received GC therapy. The complete remission rates were 50% and 54% in the CYC and MMF groups, respectively. Gastrointestinal symptoms were significantly more frequent in the MMF-treated patients, but other adverse events were similar [202]. Although gastrointestinal symptoms can occur with MMF and CYC, MMF symptoms tend to be mild and self-limiting, while in the CYC group, the risk of dehydration, hospitalizations, and discontinuation of therapy is higher. MMF is preferred in young men and women due to the high risk of sperm abnormalities and gonadal failure associated with CYC [203,204]. In a retrospective analysis of 63 patients, the ALMS and AURA clinical trials compared high- and low-dose MMF treatment combined with GCs in LN. The low-dose regimen showed no decrease in efficacy, but reduced the risk of lymphoproliferative disorders, skin cancer, and GC-related side effects [205].

As maintenance therapies, MMF and AZA are more effective and less toxic than CYC [17]. In a clinical trial that included 227 patients, treatment failure rates were 16.4% and 32.4% in the MMF and AZA groups, respectively. While minor side effects, such as infections and gastrointestinal disorders, occurred in more than 95% of patients in both groups, serious adverse effects occurred in 33.3% and 23.5% of the AZA and MMF groups, respectively [206]. MMF is the most widely used agent in maintenance treatment. However, MMF’s superiority over AZA was neither affirmed in the MAINTAIN study that compared them in the long term [207], nor in a meta-analysis study that included seven controlled clinical trials, where no significant differences were found between groups in terms of mortality, relapse, exacerbation of renal disease, doubling of serum creatinine, infection, or gastrointestinal symptoms. Nonetheless, the MMF group had a lower risk of leukopenia and amenorrhea [203].

MMF’s efficacy in managing non-renal SLE manifestations has only been published in case reports and uncontrolled clinical trials. In a retrospective study where a cohort of patients with vasculitis and SLE were treated with MMF and GC, the therapeutic GC doses were significantly lower when combined with MMF, and 46% of the patients responded well to therapy [208]. In their systematic review published in 2017, Fong et al. suggested that MMF may be efficacious in managing refractory SLE manifestations, such as hemolytic anemia, thrombocytopenia, and cutaneous lupus, and patients with low-grade disease activity that is irresponsive to other immunosuppressive agents, such as AZA and MTX [209]. There are reports of the use of MMF in treating pulmonary hemorrhage, interstitial lung disease, pericarditis, and myocarditis [210]. However, it is not possible to definitively determine the optimal dose or duration of treatment, as more compelling studies are required to make recommendations.

### 6.3. Safety

Patients treated with MMF have a lower risk of ovarian failure, alopecia, leukopenia, and serious infections compared to CYC, but diarrhea is more common with MMF [211]. Gastrointestinal symptoms, such as diarrhea, nausea, and vomiting, are more frequent with peak plasma concentrations of MPA. For better tolerance, it is advisable to subdivide the daily MMF dose into two or three administrations [139]. MPS has a lower peak plasma concentration and may decrease the incidence of gastrointestinal events compared to MMF [139].

The risk of leukopenia is low with the doses used in treating SLE [2]; hence, serious infections occur in less than 12%, and herpes zoster infection occurs in 4–18% of patients exposed to MMF [212]. Cases of progressive multifocal leukoencephalopathy (PML) have been reported in MMF-treated SLE patients [213], with hemiparesis, apathy, confusion, cognitive alterations, and ataxia being the most frequent manifestations [213].

### 6.4. Safety in Pregnancy

MMF is contraindicated in pregnancy because about 45–49% of MMF-treated pregnant women go through spontaneous abortions, and 23–27% of babies show congenital malformations, such as cleft lip, cleft palate, microtia, external auditory canal atresia, micrognathia, coloboma, and hypertelorism. Other less frequently reported anomalies are abnormalities in the extremities, congenital heart defects, esophageal atresia, diaphragmatic hernia, vertebral defects, and renal anomalies [214]. Due to its speculated teratogenic potential, the European Medicines Agency recommends that sexually active men on MMF use a condom during sexual intercourse and for 90 days after therapy discontinuation. Additionally, it recommends that men donate sperm no earlier than 90 days after MMF treatment [214]. Furthermore, breastfeeding is not recommended during MMF treatment; however, there is only tenuous evidence for this recommendation. Rats secrete MPA in their breast milk, but there is no data for human breast milk [214,215].

### 6.5. Monitoring

The following recommendations have been found to maximize effectiveness, while minimizing side effects using MPAs.

There is pharmacokinetic variability with MPA metabolism, and side effects are more probable with higher plasma concentrations in SLE patients. Hence, ascertaining the MPA concentration per patient can help reduce the risk of adverse reactions and improve effectiveness. Plasma MPA levels can be requested before and after any modification in MPA therapy, or when initiating or stopping concomitant medications [212].Monitor patients with previous hepatitis B virus (HBV) or hepatitis C virus (HCV) infection for signs of reactivation [216].Complete Blood Count weekly for the first month, twice a month for the second and third months, and then monthly for the first year of therapy [216].Watch for signs/symptoms of infection [216].Perform a pregnancy test 8 to 10 days before starting MMF, another immediately before starting the drug, and periodically with controls [217].Women must use two effective contraception methods simultaneously before starting and during treatment and for six weeks after treatment [217].Sexually active men (including vasectomized ones) taking MMF are advised to use condoms for intercourse during treatment and for 90 days after cessation. Their partners of childbearing potential should also use contraception during the same period [217].Patients should be advised not to donate blood during therapy or within six weeks of stopping treatment [217].Men should not donate sperm during therapy or for 90 days after discontinuation [217].

## 7. Calcineurin Inhibitors

### 7.1. Tacrolimus

Tacrolimus (TAC, FK506) is an immunosuppressant macrolide isolated from *Streptomyces tsukubaensis* in a soil sample obtained from Tsukuba, Japan, in 1984 [173,218,219]. It is categorized as a calcineurin inhibitor (CNI), along with its predecessor, cyclosporin A (CSA). However, TAC is more potent (about 30–100 times) and less toxic than CSA [220,221]. It was initially employed in preventing allograft rejection in transplant patients with very satisfactory outcomes [219]. Later, several studies demonstrated its efficacy in managing autoimmune diseases, such as SLE, rheumatoid arthritis, and psoriasis [222,223,224]. TAC is currently used in treating SLE, particularly recalcitrant LN, severe cutaneous and discoid manifestations [62,188,218].

#### 7.1.1. Mechanism of Action

Calcineurin is a calcium/calmodulin-dependent serine/threonine protein phosphatase involved in T-cell activation. When activated, calcineurin dephosphorylates and activates target transcription factors, chiefly, nuclear factor of activated T-lymphocytes (NFATs), crucial in IL-2 expression and T-cell activation. TAC complexes FK506 binding protein 12 (FKBP12) and inhibits calcineurin’s function by binding it tightly in the cytosol. Consequently, NFAT is left inactive and unable to upregulate IL-2 transcription. Hence, T-cell activation and subsequent secretion of cytokines, such as IL-2, IL-1β, IFN-γ, TNF-α, IL-6, and IL-10, as well as B-cell activation and antibody class-switching, is impaired [188,218,225,226].

Also, IL-2 production can be attenuated by inhibiting nuclear factor κβ (NF-κβ). TAC favors Iκβ/NF-κβ complex formation (the inactive state of NF-κβ), which leads to modulation of pro-inflammatory gene expression [227,228]. TAC has also been associated with other actions in vitro, such as promoting the expression of transforming growth factor-beta 1 (TGF-β1), which could underly the nephrotoxicity and pulmonary fibrosis associated with this drug [229].

#### 7.1.2. Efficacy

Tacrolimus is available in injection, capsules, or ointment formulations for intravenous, oral, or topical administration, respectively [230]. TAC is strongly recommended for treating SLE, especially recalcitrant LN and severe cutaneous manifestations, due to its top-grade efficacy and safety profile [188,227,231]. Topical formulations of TAC have excellent effectiveness in treating a wide range of cutaneous autoimmune disease manifestations, such as atopic dermatitis, psoriasis, perineal Crohn disease, uveitis, and SLE [222,223,224,227]. Cutaneous autoimmune lesions are widespread and often mar its victims [231]. These cutaneous diseases are mainly treated with systemic immunosuppressants, such as HCQ, MMF, CYC, AZA, or methotrexate, with some good outcomes. However, this approach is fraught with systemic side effects stemming from the generalized immunosuppression [231,232]. Better results are achieved when topical TAC is employed, and the systemic side effects are circumvented [231]. Lampropoulos et al. showed that even 0.1% topical TAC was efficacious in treating cutaneous SLE resistant to other treatments [231].

In LN mouse models, TAC diminished proteinuria and preserved their renal function by stabilizing podocyte cytoskeleton and preventing podocyte apoptosis [218]. In these models, TAC also suppressed the progression of glomerular hypercellularity, crescent formation, and serum anti-dsDNA antibody levels [188]. Thus, it has been used in humans as both induction and maintenance therapy for LN, usually in combination with GCs (“duo therapy”) or with MMF added (“triple therapy”) [188]. Some small randomized clinical trials (RCTs) show that TAC is as potent as intravenous CYC in treating proliferative LN, but a meta-analysis study suggests that it is superior to CYC as induction therapy in LN. However, some of these reports should be taken with some reservations as their sample sizes were small [218]. RCTs comparing TAC with MMF and CYC as induction therapy in proliferative and membranous LN or with AZA as maintenance therapy showed equal efficacies in all cases [218]. Even in LN classes III, IV, and V, TAC (0.06–0.1 mg/kg/day) was shown to be non-inferior to MMF (2–3 g/day) over 6 months in an RCT involving 150 patients [218].

TAC combined with MMF (triple therapy) is even more potent. Low-dose MMF-TAC combination was shown in an RCT to be superior to IV CYC pulses over 24 weeks (45.9% versus 25.6%; *p* < 0.001) [233]. The same regimen was successfully used to treat recalcitrant LN and patients from diverse backgrounds (African Americans and Caucasians) with proliferative LN partially responsive to MMF treatment [218].

However, it is noteworthy that long-term evidence about the effectiveness of TAC is lacking, as most RCTs only extend to six months [188,218]. Secondly, the CNIs (TAC and CSA) exhibit inter- and intra-individual pharmacokinetic variability due to inherent high variability in absorption, distribution, metabolism, and clearance [234]. Lastly, TAC shows superior efficacy in Asians with LN than in other subgroups [188].

#### 7.1.3. Safety

##### Drug–Drug Interactions

TAC undergoes substantial first-pass metabolism, mediated by hepatic cytochrome P450 enzymes, CYP3A4, and CYP3A5. Certain CYP3A5 polymorphisms are associated with increased TAC clearance and others with slower clearance. Hence, for optimal efficacy, the CYP3A5 genotyping of TAC-treated patients should be determined, as it would have dosage and toxicity implications [173,230,235]. TAC is also metabolized by P-glycoprotein (P-gp), whose expression levels are thought to be a good predictor of the dose requirements, especially within the first week of transplant [173,230,235]. TAC is often used concurrently with other drugs that are essential for transplant patients. Some of these drugs, including ketoconazole, cyclosporine A, diltiazem, erythromycin, and fluconazole, are also metabolized by P-gp and the CYP3A enzymes; hence, they affect the metabolism of TAC, decreasing its clearance [230,236]. Conversely, Rifampicin potentiates the elimination of TAC, reducing its bioavailability. Therefore, these drug-drug interactions should be considered in TAC dosage determination [230,236].

##### Renal Effects

Nephrotoxicity associated with CNIs is the primary concern with their usage; hence, it is dubbed their “Achilles heel” [237]. CNIs induce vasoconstriction of the afferent renal arteriole by elevating vasoconstrictors, such as endothelin and thromboxane, and activating the renin-angiotensin system, while suppressing vasodilator factors, such as prostaglandin E2, prostacyclin, and nitric oxide. In addition, CNIs inhibit COX-2, which contributes to the vasoconstriction, resulting in a reduced glomerular filtration rate [234,237]. This reversible, hemodynamically mediated renal dysfunction is known as “acute CNI nephrotoxicity” and is reversible [234,237]. Moreover, free radical formation plays a role in acute nephrotoxicity [237]. In addition to the hemodynamic effects, CNIs cause renal tubular functional alterations, leading to hypomagnesemia, hyperkalemia, hyperuricemia, and hyperchloremic metabolic acidosis [237]. Acute nephrotoxicity is associated with high systemic CNI doses [218].

In 1984, Myers et al. found that in addition to acute nephrotoxicity, long-term use of CSA in heart transplant recipients was associated with irreversible renal functional deterioration due to irreversible and progressive tubulointerstitial injury and glomerulosclerosis [238]. This irreversible injury was also found with TAC and termed “chronic CNI nephrotoxicity” [237,239]. Histological features, including arteriolar hyalinosis, tubular atrophy, interstitial fibrosis, and focal segmental or global glomerular sclerosis, are typical, but not pathognomonic of chronic CNI nephrotoxicity [218]. However, TAC appears to be less renotoxic than CSA because of its weaker vasoconstrictive effect and lower fibrogenic potential [218,237]. Risk factors for CNI nephrotoxicity include higher doses, concurrent use of other nephrotoxic drugs (e.g., NSAIDs), salt-depleting medicines and diuretics, and older kidney age. Others include genetic polymorphisms of the liver cytochrome enzymes (CYP3A4/5) and the multidrug efflux transporter P-gp and interactions with CYP3A4 and P-gp inhibitor drugs (e.g., ketoconazole) [218,234,237].

##### Neurological Effects

TAC-induced neurotoxicity occurs in approximately 25–31% of treated patients, of which 20% experience mild symptoms, such as tremors (most common), headaches, vertigo, photophobia, dysesthesia, paresthesia, mood disturbances, and insomnia [240]. Major neurotoxic symptoms, including confusion, seizures, cortical blindness, encephalopathy, and coma, occur in 5–8% of treated patients [240,241,242]. The major complications usually manifest within 30 days and are linked with high plasma TAC levels [241]. Rare complications, such as peripheral neuropathy and posterior cerebral edema syndrome, may occasionally occur [240,243]. These neurologic symptoms are reversible with cessation of TAC administration [240,241,243]. Some risk factors for developing TAC neurotoxicity include intravenous administration, high blood levels, and concurrent use with CYP3A4 inhibitor drugs (e.g., Nefazodone) [240].

##### Metabolic Abnormalities

Treatment with the CNIs is linked to metabolic disorders, including hyperglycemia, hyperuricemia, hypomagnesemia, and hyperkalemia, which are alterations that have been described in patients taking TAC; however, they tend to be less frequent compared to the other calcineurin inhibitors [244,245]. While hyperlipidemia (elevated LDL-cholesterol and triglyceride levels) is more uncommon with TAC treatment than CSA, diabetes mellitus (DM) is more frequent with TAC treatment [188,244]. TAC was shown to reversibly inhibit insulin mRNA transcription, insulin synthesis, and ergo insulin secretion in both in vitro and in vivo studies [244,246]. TAC-affected islet beta-cells show degranulation, vacuolation, and swelling of the mitochondria, Golgi apparatus, and rough endoplasmic reticulum [247]. These abnormalities usually resolve with dose reduction, although sometimes they require treatment suspension [244]. Consequently, all CNI-treated patients should have serum glucose, lipid profile, serum uric acid, and electrolyte monitoring [222].

##### Infections

Relative to other immunosuppressive regimens for SLE patients, TAC has a top-grade efficacy-to-toxicity ratio. In a comparative meta-analysis study of several immunosuppressive drugs for SLE, TAC was associated with a significantly lower risk of severe infections than AZA, MMF, GC, and CYC, only matched by MMF-AZA combination therapy [183]. Nevertheless, gram-negative sepsis and cytomegalovirus infection, as well as herpes simplex virus and chickenpox infections, have been described in transplant patients [183,248]. The risk of infections is linked with the concomitant use of other immunosuppressants, such as AZA, and hematological disorders, such as leukopenia [249]. Hence, clinical and laboratory monitoring is recommended for patients on CNIs [249].

#### 7.1.4. Safety in Pregnancy and Lactation

Unlike CYC and MMF, TAC (along with AZA) is one of the few pregnancy-compatible immunosuppressants for SLE patients because it has no ill effects on fertility in women [214,250,251]. Albeit AZA is considered the first-choice medicine for pregnant LN patients, TAC is indicated in AZA-resistant or AZA-intolerant cases [188,218]. Further, only a negligible amount of TAC is detectable in breastmilk; hence, it is breastfeeding-safe and recommended for younger patients who want to preserve fertility [218]. Table 6 summarizes organ-specific side effects Tacrolimus therapy.

#### 7.1.5. Recommendations in Drug Administration and Monitoring

There are two recommended administration forms: the conventional and modified release forms. The latter attempts to favor the drug’s bioavailability and, therefore, reduces the dosage to once daily, increasing patient adherence to the treatment. Ideally, it should be taken on an empty stomach, and it is not recommended for the patient to chew the drug [173].

Monitor frequently: blood glucose, renal function, liver function, serum potassium levels (especially in patients receiving other medications associated with hyperkalemia), electrolytes (i.e., magnesium, potassium, calcium) [245,252].Monitor ECGs periodically during treatment, especially in patients at risk for QT prolongation (concomitant use of other QT-prolonging drugs or CYP3A inhibitors, electrolyte disturbances, congestive heart failure, or bradyarrhythmia) [253].Surveillance for signs/symptoms of opportunistic infections.Surveillance for signs/symptoms of Neurologic abnormalities.

### 7.2. Cyclosporine

Cyclosporin A (CSA) was introduced as an alternative immunosuppressant in 1980 [254]. It is a calcineurin inhibitor that preferentially binds to cyclophilin, unlike tacrolimus, which has an effect on FKBP12. Cyclosporine is generally considered to have a lower potency (up to 100-fold less) than TAC [250].

Inhibition of calcineurin prevents the translocation of cytokine-related transcription factors (such as those responsive to IL-2), with subsequent inactivation of T cells achieving modulation of autoimmune activity [255]. In addition, cyclosporine has been shown to have effects on podocytes that may reduce proteinuria [250,254,256].

CSA is a lipophilic drug, with a narrow therapeutic range. It is metabolized mainly through CYP3A4 and is a substrate of P-glycoprotein. Its pharmacokinetics can be altered by food intake or even in situations such as hypoalbuminemia or hepatic failure [254,256].

#### 7.2.1. Safety

In general, calcineurin inhibitors have been associated with metabolic, hematological, renal, and neurological effects. Monitoring of cyclosporine levels is recommended to avoid toxicity-associated effects [256]. Adverse effects usually improve after discontinuation of the drug. Currently, in different consensuses, the use of TAC is preferred over CSA due to its better safety profile and better control of the disease during maintenance therapy [17,256].

#### Nephrotoxicity

Acute and chronic nephrotoxicity have been described with cyclosporine. The drug’s own vasoconstriction and fibrogenic potential were considered as factors associated with the development of this adverse reaction. The reduction in glomerular filtration rate associated with these drugs has not been associated only with elevated levels—it appears that other associated pathological or genetic conditions could trigger renal injury. It is recommended to monitor serum electrolytes as cyclosporine may be associated with hyperkalemia [256].

#### Metabolic

Cyclosporine patients are at risk of developing dyslipidemia and hirsutism, as well as hypertension. Monitoring of blood pressure, lipid profile, and changes in body hair distribution is recommended [256].

## 8. Methotrexate (MTX)

MTX is an antifolate drug derived from aminopterin. It is indicated as a disease-modifying drug in RA [257,258], inflammatory polyarthritis [259], severe psoriasis [260,261], psoriatic arthritis [261], juvenile idiopathic arthritis [257], ankylosing spondylitis [262], dermatomyositis [263], polymyositis [263], Crohn’s disease [264], and SLE [258,265]. In SLE, its use is indicated in patients who respond inadequately to antimalarials [266] and in patients with moderate SLE with skin, joint, and serous involvement, but without kidney involvement [258,265].

### 8.1. Mechanism of Action

MTX enters cells through the folate transporter type [257] and the reduced folate carrier type 1 (RFC1) [267]. At the intracellular level, MTX in the form of monoglutamate undergoes glutamic acid additions, forming polyglutamates, a more active and potent form of drug [267]. These polyglutamates inhibit several enzymes: (a) 5-aminoimidazole-4-carboxamide ribonucleotide (AICAR) transformylase (ATIC), an enzyme that participates in the de novo biosynthesis of purines [267]—when this enzyme is inhibited, levels of adenosine, a molecule with anti-inflammatory effects, increase [262]; (b) thymidylate synthase (TYMS), an enzyme that participates in the synthesis of pyridimines; (c) enzymes involved in polyamine transmethylation and synthesis reactions, thus decreasing the production of ammonium and H_2_O_2_, harmful agents for cells and joint tissues [267]; (d) dihydrofolate reductase (DHFR) and methylenetetrahydrofolate reductase (MTHFR), folate-dependent enzymes involved in the synthesis of purines, thymidylate, serine, methionine, and DNA [257,262].

Inhibition of DHFR further inhibits the production of tetrahydrobiopterin (BH4), a cofactor of nitric oxide synthase. By decreasing the activity of this enzyme, the formation of nitric oxide is decreased and the production of ROS that activate JUN N-terminal kinases (JNKs) increases. This activation increases the activity of the transcription factors AP-1 and NF-κB, promoting apoptosis of inflammatory cells [267].

### 8.2. Efficacy

In rheumatic diseases, MTX is administered once a week, orally, subcutaneously or intramuscularly [268]. It can be considered in moderate or severe SLE [268], which responds sub-optimally to antimalarials and in those cases where it is not possible to reduce the GC dose [269]. It is administered in doses of 7.5 to 25 mg per week, and favorable responses can be observed in 4 to 12 weeks [268]. Parenteral MTX seems to be useful in general, especially in those patients with insufficient response to oral MTX [270]. The parenteral route does not seem to increase the rate or severity of adverse events compared to the oral route and could reduce costs in those patients with an inadequate response to oral MTX [270].

Sakthiswary et al. evaluated the evidence for the use of MTX in SLE in a systematic review that included three controlled trials [258,271,272] and five observational studies [273,274,275,276,277], finding a significant reduction in the SLEDAI score and reduction in the mean dose of GC among patients treated with MTX [269]. MTX also reduces the average use time of prednisone [258], and this GC-sparing effect is relevant in light of the risk of adverse reactions associated with its use [258].

Patients with SLE where there is evidence of benefit are those with joint, cutaneous, and serous involvement [268,272,274]. In contrast, it seems to be effective in improving the serological alterations that are frequently observed during a lupus flare, observing increased levels of C3 and C4 and a decrease in the levels of anti-dsDNA, IgG, IgA, and IgM antibodies [277]. It is not recommended for interstitial lung disease, hepatitis, or cytopenias [268], and clinical trials where its efficacy has been studied have excluded patients with LN and NPSLE [278].

### 8.3. Safety

Although it is generally well tolerated, the use of MTX can cause pancytopenia, hepatotoxicity, pulmonary toxicity, nephrotoxicity, gastrointestinal adverse events, and skin rashes [279]. The most frequently reported adverse events are gastrointestinal [67,280] and leukopenia [67]. The prevalence of adverse events varies between 10–70% of patients with SLE [220,278,280], leading to the suspension of treatment in 19–33% of cases [67,280].

#### 8.3.1. Gastrointestinal Side Effects

The prevalence of gastrointestinal symptoms varies between 20% and 40% in patients taking MTX [221]. The most frequent manifestations are nausea, vomiting, abdominal pain, and mucositis [281]. Risk factors for its presentation include doses greater than 8 mg/week [221]; concomitant drugs, such as NSAIDs; bisphosphonates and GC [221]; the absence of folic acid supplementation [282]; and kidney disease [279].

To reduce the risk of gastrointestinal reactions, it is recommended: (a) consider the administration of MTX as a relative contraindication in patients with active gastric ulcer; (b) supplement with folic or folinic acid at a dose greater than 5 mg/week [282,283]; (c) start with doses of 12.5 to 20 mg/week and slowly titrate [283]; (d) administer divided oral doses of MTX [284]; and (e) if gastrointestinal symptoms persist, consider changing the route of administration of MTX from oral to parenteral [270,283,284].

#### 8.3.2. Hepatotoxicity

MTX can cause elevation in liver function tests in 10–43% of patients [285]. Temporary suspension of the drug or dose adjustment in general produces resolution of these alterations; however, the evolution to chronic disease due to fibrosis has been described [281,285]. Risk factors associated with hepatotoxicity include alcohol consumption, obesity, hypercholesterolemia, elevation of liver function tests before starting treatment with MTX, use of biological agents, absence of folic acid [286], advanced age, hypoalbuminemia, diabetes mellitus, kidney failure, and viral hepatitis [285].

Pathological changes found in liver biopsies include hepatic steatosis, focal necrosis, liver fibrosis, chronic inflammatory infiltrate in portal tracts, and nuclear pleomorphism [287].

#### 8.3.3. Hematological Side Effects

Between 1–12% of patients treated with MTX present cytopenias [285,288], and in up to 1.4%, pancytopenia [288]. Risk factors for the development of pancytopenia include: advanced age, renal failure, hypoalbuminaemia, daily intake of MTX due to medication error, absence of folic acid substitution, polypharmacy [288], and being a carrier of the C677T polymorphism of methylenetetrahydrofolate reductase (MTHFR) [289].

The temporary suspension of the drug recovers the moderate suppression of the bone marrow within two weeks after withdrawal. However, a mortality rate between 17% and 44% can occur in patients with pancytopenia secondary to sepsis [288].

In contrast, MTX treatment has also been associated with lymphoproliferative disorders. Associated risk factors include intense immunosuppression, genetic predisposition, and increased frequency of latent infections with pro-oncogenic viruses [285,290,291].

#### 8.3.4. Pulmonary Side Effects

Approximately 1% to 8% of patients receiving treatment with MTX may present with pulmonary alterations, such as interstitial pneumonitis [285,291]. Its presentation is independent of the accumulated dose and the duration of treatment [285]. Risk factors for the presence of MTX-induced pneumonitis include age older than 60 years, diabetes mellitus, hypoalbuminemia, previous use of DMARDs, kidney dysfunction, male gender, and pre-existing lung disease [292].

The proposed mechanism, although not clear, include hypersensitivity, direct toxicity of the drug, and repeated viral infections [292]. Within the paraclinical findings, it is possible to find a restrictive pattern in pulmonary function tests; a diffuse interstitial pattern on chest radiograph; ground glass opacities, with or without consolidation foci [291]; and basal fibrosis on CT in more advanced stages [292].

#### 8.3.5. Renal Side Effects

The etiology of MTX-induced renal dysfunction is mediated by a direct toxic effect or by precipitation of MTX and its metabolites in the renal tubules at acidic urinary pH [293]. This crystallization generates infiltration of inflammatory cells and oxidative stress at the level of the renal tubules, which manifests with an increase in renal function tests and greater deterioration in the excretion of MTX [294].

Nephrotoxicity with low doses of MTX can be precipitated by doses not adjusted to renal function or by concomitant treatment with drugs that interfere with the excretion of MTX, such as probenecid, salicylates, sulfisoxazole, penicillins, and non-steroidal anti-inflammatory agents [293].

#### 8.3.6. Neurotoxicity

The neurotoxicity induced by MTX is described mainly in patients who receive the drug at high doses or intrathecally [295]. The manifestations described include acute, subacute, or chronic neurotoxicity [296]. Acute neurotoxicity occurs after hours of administration of the drug and includes drowsiness, disorientation, seizures, headache, and dizziness. Subacute toxicity presents after days to weeks of treatment and includes findings of encephalopathy or myelopathy. Finally, chronic neurotoxicity, which occurs months to years of treatment, can manifest with cognitive alterations, dementia, and leukoencephalopathy [295].

At low doses, neurological symptoms are infrequent, mainly dizziness, vertigo, or headache [297].

### 8.4. Safety in Pregnancy

Associated abnormalities include spontaneous abortions, preterm delivery, metatarsal varus, palpebral angioma, growth deficiency, dysmorphic facies, multiple skeletal abnormalities of the skull and extremities, and less frequently, central nervous system abnormalities and congenital heart defects [298].

### 8.5. Monitoring

The following recommendations should be followed to ensure minimal side effects and optimal selection of candidates for MTX therapy.

Evaluate risk factors for serious adverse events due to MTX, such as alcohol intake, age over 70 years, and acute or chronic infections [299]. Avoid starting the drug in these patients [299].Perform a complete blood count before the start of treatment and at least once a month during the first 3 months. It should then be done every 4 to 12 weeks during therapy [283,285].Perform liver function tests before the start of treatment, every month for the first 3 months, and then every 2 to 4 months [283,285].If liver function tests are elevated less than three times the upper normal value, a dose reduction is recommended. If they are persistently elevated more than three times the upper normal value despite dose reduction, it will be necessary to suspend the drug [299] and carry out complementary studies with evaluation by hepatology if the elevation of transaminases persists despite suspension [299].It is recommended to take hepatitis B and C serology and measure serum albumin before starting treatment and repeat it in those patients who persist with altered liver function tests despite suspension of treatment [285].The patient should receive simultaneous treatment with folic acid to reduce the adverse events associated with treatment with MTX [282,284,285].Perform a pregnancy test before the start of treatment and periodically during treatment. Discuss with the patient the importance of contraception during treatment and the need to discontinue treatment with MTX 3 months before conception [299].Determine the glomerular filtration rate before starting treatment, every month for the first 3 months, and then every 4–12 weeks during treatment [285]. The dose of MTX should be adjusted to renal function: Glomerular filtration rates (GFR) between 30 and 60 mL/min, require a reduction of the MTX dose of 30–50%, and perform renal function tests during therapy, initially twice a week and then every 4 weeks. The administration of MTX with a GFR <30 mL/min is not recommended [294].Evaluation of respiratory symptoms and history in patients with suspected parenchymal lung disease, perform pulmonary function tests and chest radiography. Consider more frequent monitoring of respiratory symptoms and pulmonary function tests during therapy in this type of patient [299].

## 9. Dapsone

Dapsone, or 4,4-diaminodiphenylsulfone, is currently considered second-line therapy in bullous systemic lupus erythematosus (BSLE), either in monotherapy or in combination with GC [300,301,302]. Additionally, it is a treatment option in some refractory types of cutaneous lupus erythematosus (CLE), such as discoid lupus erythematosus (DEL) and subacute cutaneous lupus erythematosus (SCLE) [303].

After oral administration, it has a bioavailability of 70–80% [304]. It is hepatically metabolized by hydroxylation (CYP2E1, CYP2C9, CYP3A4) to dapsone hydroxylamine (DDS-NOH) or by N-acetylation to monoacetyldapsone (MADS) [305]. The parent molecule and its metabolites are conjugated with glucuronic acid or sulfate for renal elimination [304]. It has a volume of distribution of 1.5 L/kg, reaching most of the tissues, especially skin, kidney, liver, central nervous system, and placenta [305].

### 9.1. Mechanism of Action

The anti-inflammatory mechanism of action of dapsone involves multiple pathways: inhibition of pro-inflammatory cytokines, such as IL-8 [305,306]; alteration of chemotaxis and integrin-mediated neutrophil adhesion [307]; inhibition of leukocyte and eosinophil myeloperoxidases enzymes [308]; decrease in the generation of reactive oxygen species; and inhibition of the arachidonic acid cascade, thereby decreasing the generation of 5-lipoxygenase, prostaglandin E2, and thromboxane products [302].

### 9.2. Efficacy

EULAR recommends the use of dapsone at a dose of 100 mg per day in patients with BSLE who do not respond to or require high doses of GC [309]. However, it is common to start with a dose of 50 mg per day, which is titrated according to response and tolerance up to a maximum of 200 mg per day [300,309].

BSLE occurs in less than 5% of SLE cases [300] and may be the initial manifestation of the disease or be associated with lupus activity, in which case bullous lesions occur more frequently with lupus nephritis [300]. Most of the evidence for the use of dapsone, due to the frequency of the disease, results from case reports and retrospective analyses [300,310,311,312].

Hall et al. reported four patients with GC-resistant BSLE who, within the first day of dapsone therapy, had improvement of the lesions [310].

Lourenço et al. report 3 cases of BSLE in children aged 5 to 10 years. Two were treated with dapsone with improvement of the lesions on average between four weeks and four months of treatment. No adverse reactions were reported [313].

In a retrospective analysis of 181 cases of patients with BSLE, 91% of patients treated with dapsone improved partially or completely; however, treatment was discontinued in 23% of patients due to adverse reactions, mainly anemia, hypersensitivity reactions, and hepatitis [314]. Discontinuation of dapsone therapy before one year may result in recurrence of lesions, but they respond to reintroduction of the drug [300].

### 9.3. Safety

Hematological, cutaneous, and immunological, neuropsychiatric, gastrointestinal and hepatic alterations may occur [305].

#### 9.3.1. Hematological Alterations

DDS-NOH, being a potent oxidizing agent, can submit the erythrocyte to oxidative stress, inducing hemolytic anemia, and can also oxidize the iron in hemoglobin, generating methemoglobinemia [305]. Some risk factors for these alterations include high doses, pre-existing hemoglobin abnormalities, low levels of cytochrome b5 reductase enzyme activity, glucose 6 phosphate dehydrogenase (G6PD) deficiency, and the use of other drugs that may induce methemoglobinemia [315]. Small amounts of DDS-NOH could be transported by the erythrocytes to the bone marrow, where it could possibly interact with neutrophils and induce agranulocytosis [305].

#### 9.3.2. Cutaneous and Immunological Alterations

Dapsone has been associated with several cutaneous adverse reactions, such as fixed rash, exfoliative dermatitis, erythema nodosum, erythema multiforme, morbilliform and scarlatiniform rashes, Stevens–Johnson syndrome, toxic epidermal necrolysis, and DRESS [305,316].

Dapsone hypersensitivity syndrome (DHS), an idiosyncratic adverse reaction with multiorgan involvement, has been described [317], which develops within 1 to 6 weeks after the start of treatment [318], but it could occur even in the first 6 h of exposure in a previously sensitized individual [317], or after six months of therapy [319]. It has an incidence of 1.4% [320], and among the most frequent manifestations are fever, skin lesions, hepatosplenomegaly [317], hepatic lesion with a more frequent cholestatic than hepatocellular pattern, lymphadenopathy, nausea, vomiting, mucosal involvement, hematological alterations (hemolysis, agranulocytosis, leukocytosis, anemia, eosinophilia, reticulocytosis, atypical lymphocytosis, leukemoid reaction [317], interstitial pneumonitis, carditis, and nephritis [319].

Duration of DHS is about four weeks or more, usually self-limiting with drug discontinuation, but systemic GCs are often used as adjuvants [317]. A mortality rate of 11–13% has been described [315,318].

#### 9.3.3. Neuropsychiatric

Ischemic optic neuropathy [315] and motor-predominant axonal degenerative peripheral neuropathy have been described. They may improve after one year of drug withdrawal, but recovery may be delayed and partial [302]. In addition, psychiatric symptoms, including irritability, insomnia, and confusion have been reported [321].

#### 9.3.4. Gastrointestinal

Reactions at this level include nausea, vomiting, abdominal pain, liver injury, and pancreatitis [302,322,323]. The highly reactive metabolite DDS-NOH induces oxidative stress and lipid peroxidation in the liver, leading to hepatic necrosis, hepatitis, and cholestasis [322,323].

Cases of pancreatitis have been reported within 4 months after initiation of the drug, or weeks after dose increase in patients on prolonged therapy. All cases have resolved with discontinuation of the drug [323].

#### 9.3.5. Safety in Pregnancy and Lactation

Dapsone is considered category C in pregnancy because it crosses the placental barrier, but no teratogenic effects have been observed in animals or humans [324].

It can be administered during lactation. However, because it is eliminated through breast milk, it should be avoided in infants with G6PD deficiency and/or hyperbilirubinemia [324].

### 9.4. Monitoring

Avoid use in patients with a history of allergy to sulfas and in patients with severe liver disease [302].Determine glucose-6-phosphate dehydrogenase levels prior to initiating therapy [302,305].Perform baseline CBC, then weekly for the first month, monthly for 6 months, and then semi-annually thereafter [305].Request reticulocyte count at the beginning of treatment, and then periodically every 3–4 months [305].Perform liver and renal function tests at the start of treatment, and then every 3–4 months thereafter [305].Consider determining the methemoglobin level at the beginning of treatment and according to symptoms [302].Monitor for clinical signs of jaundice, hemolysis, and blood dyscrasias during each visit [302,305], inquire about adverse reactions, monitor for neurological and psychiatric disorders [302].

## 10. Conclusions

SLE is one of the most common autoimmune diseases affecting our modern societies, hence, several immunomodulatory or immunosuppressive drugs, including antimalarials and glucocorticoids, have been developed to manage the disease. In this review article, we described current therapies and their possible side effects.

Having been in clinical use for several decades, the antimalarials have been rigorously studied. They are immunomodulatory rather than immunosuppressive; hence, their usage is associated with less risk of infection and cancer, and they are better tolerated than other treatment alternatives. Besides their high efficacy, antimalarials are also considerably safer than many other SLE drugs, as their side effects tend to be mild, few, and rare, and they are among the very few SLE drugs not contraindicated during pregnancy. However, they are generally employed in symptomatic management and not useful as induction therapy. Glucocorticoids are probably the most essential drug in treating autoimmune diseases, such as SLE. Owing to their efficacy as immunosuppressants, GCs are used to manage the most severe SLE manifestations as induction therapy, but are also commonly used as maintenance therapy, usually in combination with other treatments. Albeit effective, GCs engender significant dose-dependent side effects; hence, it is good practice to taper their dosage over the shortest amount of time possible. Antimalarials and GCs are both essential drugs in the doctor’s cabinet for managing SLE. Multiple other drugs, such as cyclosporine, methotrexate, mycophenolate, azathioprine, and cyclophosphamide, are also useful in specific cases, especially when monitored carefully.

## Figures and Tables

**Figure 1 medicina-59-00056-f001:**
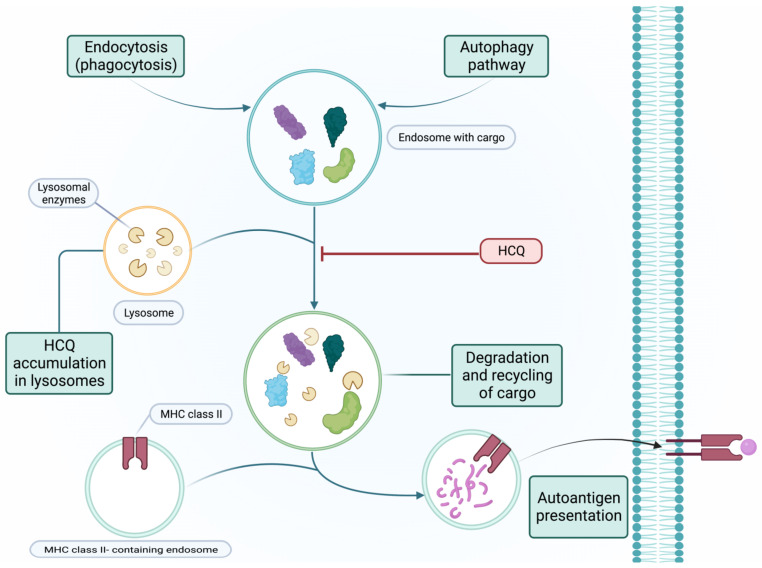
Mechanism of action of antimalarial immunomodulation during autoimmunity. HCQ accumulates in lysosomes and inhibits the degradation of cargo derived externally (via endocytosis or phagocytosis) or internally (via the autophagy pathway) in autolysosomes by increasing the pH to prevent the activity of lysosomal enzymes. Inhibition of lysosomal activity can prevent MHC class II-mediated autoantigen presentation. Adapted from Schrezenmeier et al. [6]. Created with BioRender.com (accessed on 17 October 2022).

**Figure 2 medicina-59-00056-f002:**
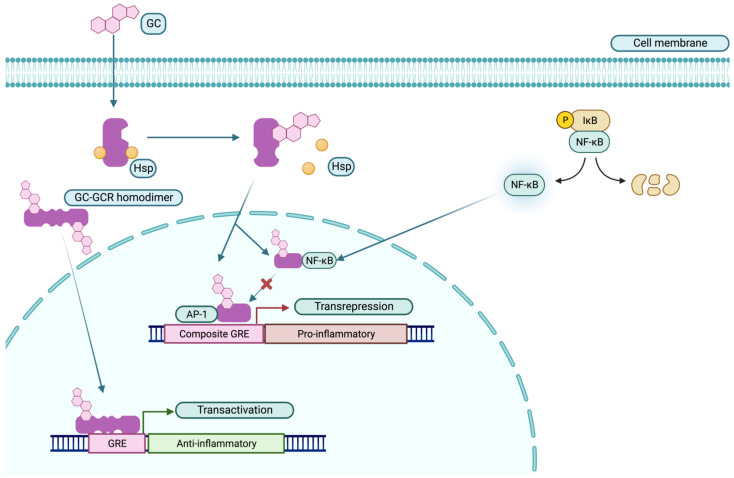
Genomic mechanisms of glucocorticoid-induced anti-inflammation. GCs bind to their cytosolic glucocorticoid receptor (GCR), which subsequently loses its chaperoning proteins, such as heat shock proteins (Hsp). Homodimers are formed, travel to the nucleus, bind to the glucocorticoid response element (GRE), and upregulate the expression of certain genes (e.g., lipocortin-1 and genes involved in metabolism), a mechanism called transactivation. Monomeric GC–GCR complex (mGC-GCR) can bind to transcription factors as AP-1 and NF-kβ, inhibiting the transcription of their target genes (e.g., IL-2 and TNFα) by a mechanism called transrepression. Further, direct binding of mGC-GCR alongside AP-1 on composite GREs lead to transrepression. Created with BioRender.com (accessed on 17 October 2022).

**Table 1 medicina-59-00056-t001:** Risk factors for antimalarial-induced ocular toxicity [5,32].

Daily HCQ dose > 5 mg/kg (real weight) CQ dose > 2.3 mg/kg (real weight) Daily HCQ dose >6.5 mg/kg of ideal weight in obese patients Use for >5 years Cumulative dose > 600–1000 g CKD stage 3, 4, or 5 Concomitant tamoxifen for >6 months Macular degeneration, retinal dystrophy, cataracts

**Table 2 medicina-59-00056-t002:** Summary of organ-specific side effects of antimalarial therapy.

Organ	Side Effects
Gastrointestinal tract	Nausea, vomiting, abdominal discomfort, diarrhea, hepatotoxicity
Skin	Pruritus (generalized, aquagenic), hyper- or depigmentation, ecchymosis, DRESS, erythema multiforme, erythroderma.
Eye	Retinopathy, diminished peripheral and nocturnal vision, bulls-eye maculopathy, difficulty reading, altered color perception.
Heart	Cardiomyopathy, bradycardia, tachycardia, T-wave flattening, left anterior fascicular block, complete atrioventricular block.
Neuromuscular system	Headaches, dizziness, insomnia, vertigo, tinnitus, hearing loss, psychosis, delirium, depression, reversible proximal myopathy, non-painful neuropathy.

**Table 3 medicina-59-00056-t003:** Summary of Organ-Specific Side Effects of High-Dose or Prolonged GC Therapy.

Organ	Side Effects
Kidney	Increased sodium retention and potassium excretion
Musculoskeletal system	Osteoporosis, osteonecrosis, myopathy and atrophy, and growth retardation,
Cardiovascular system	Dyslipidemia, hypertension, hyperglycemia, lipodystrophy and weight gain, thrombosis, and vasculitis
Adrenal gland	Adrenal atrophy and Cushing’s syndrome
Skin	Atrophy, delayed wound healing, erythema, hypertrichosis, perioral dermatitis, petechiae, glucocorticoid-induced acne, striae rubrae distensae, and telangiectasia
Eyes	Cataracts, glaucoma, myopia, exophthalmos, papilledema, chorioretinopathy, and subconjunctival hemorrhages.
Central nervous system	Depression, psychosis, bipolar disorders, delirium, panic attacks, obsessive–compulsive disorder, anxiety, insomnia, catatonia, and cognitive impairment
Gastrointestinal tract	Bleeding, pancreatitis, and peptic ulcer
Immune system	Broad immunosuppression; activation of latent viruses; increased risk of bacterial, fungal, and viral infections.
Reproductive system	Delayed puberty, fetal growth retardation, hypogonadism, gestational diabetes, hypertension, preeclampsia, premature rupture of membranes, and risk of cleft palate

Adapted from Rhen et al. [42].

**Table 4 medicina-59-00056-t004:** Summary of Organ-Specific Side Effects of Cyclophosphamide Therapy.

Organ	Side Effect
Gastrointestinal tract	Nausea, gastrointestinal dysmotility, emesis, and hepatotoxicity
Reproductive system	Ovarian failure (reduced estradiol, progesterone, maturation of oocytes, and number of ovarian follicles), amenorrhea, azoospermia, spontaneous abortions, congenital malformation, growth retardation, anatomical abnormalities, and cervical atypia
Urinary tract	Necrosis of bladder mucosa, hematuria, hemorrhagic cystitis, and bladder carcinoma
Immune system	Hematologic malignancies, neutropenia, lymphopenia, thrombocytopenia, anemia, and increased risk of infections.
Lung	Interstitial pneumonitis and fibrosis
Cardiovascular system	Hypertrophy, myocardial fibrosis, tachyarrhythmias, hypotension, heart failure, myocarditis, and perimyocarditis

**Table 5 medicina-59-00056-t005:** Summary of Organ-Specific Side Effects of Azathioprine Therapy.

Organ	Side Effect
Reproductive system	Developmental delays, pancytopenia, premature birth, mild malformations
Gastrointestinal tract	Anorexia, nausea, vomiting, and diarrhea, pancreatitis, hepatotoxicity
Immune system	Myelosuppression (leucopenia and thrombocytopenia), anemia, bleeding, increased risk of herpesvirus (CMV, VZV, HSV, EBV) and bacterial infection, non-Hodgkin’s lymphoma

**Table 6 medicina-59-00056-t006:** Summary of Organ-Specific Side Effects of Tacrolimus Therapy.

Organ	Side Effect
Kidney	Acute nephrotoxicity (hypomagnesemia, hyperkalemia, hyperuricemia, hyperchloremic metabolic acidosis), chronic nephrotoxicity (arteriolar hyalinosis, tubular atrophy, interstitial fibrosis, glomerular sclerosis)
Central nervous system	Tremor, headache, vertigo, photophobia, dysesthesia, paresthesia, mood disturbances, insomnia, confusion, seizures, cortical blindness, encephalopathy, coma, peripheral neuropathy, posterior cerebral edema syndrome
Cardiovascular system	Hyperlipidemia (high LDL cholesterol and triglycerides), hyperglycemia
Immune system	Slightly increased risk of bacteria and herpesvirus infection

## Data Availability

Not applicable.

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
