# Peer review of "Synthetic Pharmacotherapy for Systemic Lupus Erythematosus: Potential Mechanisms of Action, Efficacy, and Safety"

_medicina, 2022, doi:10.3390/medicina59010056_

Round 1
Reviewer 1 Report
An interesting paper on an important topic. A suggest several changes:
1. The abstract should be more detailed, please at least mention the current treatment options that you review in the paper
2. The paper is well-organized, however, it is lacking some kind of summary/discussion at the end
3. You should include dapsone in your review, it is used in a rare variant of bullous systemic lupus erythematosus
4. What about cyclosporin A? Are there any differences in the safety profile etc when compared to tacrolimus?
Reviewer 2 Report
Synthetic pharmacotherapy for systemic lupus erythematosus: 2 potential mechanisms of action, efficacy, and safety
Location (pages) |
Type of correction / Problem identified |
Action needed |
39-40 |
Change in text |
Where is written “...desire of children…” should be better meant by “desire of parenthood” |
83-90 |
Change in paragraph |
I recommend only one paragraph, since both paragraphs are in the same context. Paragraphs should be linked. |
83-99 |
Substitute Repetead words |
The word “Also” is fully used many times to express continuity. Possible suggestions: In addition; Furthermore; In this way; Moreover; Additionally, etc. |
103-104 |
Sentence continuity |
The sentence was abruptly cut and discontinuated, author should verify such typos |
176-198 |
Topic comprehension |
An introductory paragraph is needed in topic 2.3 Monitoring (such as was done in topic 2.2 as well). Suggestion “As observed in several references as depicted as follows, the criteria of… “ |
Minor typos in manuscript that should be checked. |
